# Apobec-mediated retroviral hypermutation *in vivo* is dependent on mouse strain

Hyewon Byun[1], Gurvani B. Singh[1¤a], Wendy Kaichun Xu[1¤b], Poulami Das[1], Alejandro Reyes[1], Anna Battenhouse[2], Dennis C. Wylie[2], Mario L. Santiago[3], Mary M. Lozano[1], Jaquelin P. Dudley🆔[1,4]*

1 Department of Molecular Biosciences, The University of Texas at Austin, Austin, Texas, United States of America, 2 Center for Biomedical Research Support, The University of Texas at Austin, Austin, Texas, United States of America, 3 Department of Medicine, University of Colorado Anschutz Medical Campus, Aurora, Colorado, United States of America, 4 LaMontagne Center for Infectious Disease, The University of Texas at Austin, Austin, Texas, United States of America

¤a Current address: Cerus Corporation, Concord, California, United States of America
¤b Current address: Genentech, South San Francisco, California, United States of America
* jdudley@austin.utexas.edu

**Data Availability Statement:** Transcriptomic and genomic data are available in NCBI GEO with accession IDs GSE263655 and GSE263741,

## Abstract

Replication of the complex retrovirus mouse mammary tumor virus (MMTV) is antagonized by murine Apobec3 (mA3), a member of the Apobec family of cytidine deaminases. We have shown that MMTV-encoded Rem protein inhibits proviral mutagenesis by the Apobec enzyme, activation-induced cytidine deaminase (AID) during viral replication in BALB/c mice. To further study the role of Rem *in vivo*, we have infected C57BL/6 (B6) mice with a superantigen-independent lymphomagenic strain of MMTV (TBLV-WT) or a mutant strain that is defective in Rem and its cleavage product Rem-CT (TBLV-SD). Compared to BALB/c, B6 mice were more susceptible to TBLV infection and tumorigenesis. Furthermore, unlike MMTV, TBLV induced T-cell tumors in B6 µMT mice, which lack membrane-bound IgM and conventional B-2 cells. At limiting viral doses, loss of Rem expression in TBLV-SD-infected B6 mice accelerated tumorigenesis compared to TBLV-WT in either wild-type B6 or AID-knockout mice. Unlike BALB/c results, high-throughput sequencing indicated that proviral G-to-A or C-to-T mutations were unchanged regardless of Rem expression in B6 tumors. However, knockout of both AID and mA3 reduced G-to-A mutations. *Ex vivo* stimulation showed higher levels of mA3 relative to AID in B6 compared to BALB/c splenocytes, and effects of agonists differed in the two strains. RNA-Seq revealed increased transcripts related to growth factor and cytokine signaling in TBLV-SD-induced tumors relative to TBLV-WT-induced tumors, consistent with another Rem function. Thus, Rem-mediated effects on tumorigenesis in B6 mice are independent of Apobec-mediated proviral hypermutation.

## Author summary

Retroviruses cause lifelong infections resulting from their ability to thwart innate immunity. The Apobec family of cytidine deaminases are part of the innate immune response that recognizes and mutates foreign nucleic acids, including those from multiple viruses.

respectively. All other relevant data are within the paper and its Supporting Information files.

**Funding:** This work was supported by Public Health Service grants R01 AI131660 (J.P.D) and R01 AI116603 (M.L.S.) from the National Institute of Allergy and Infectious Diseases (https://www.niaid.nih.gov/). The funders had no role in study design, data collection and analysis, decision to publish, or preparation of the manuscript.

**Competing interests:** The authors have declared that no competing interests exist.

Retroviral antagonists of Apobecs have been identified, including mouse mammary tumor virus (MMTV)-encoded Rem protein. Previous experiments have shown that Rem-null MMTV or closely related TBLV proviruses from BALB/c tumors accumulate G-to-A and C-to-T mutations typical of Apobecs compared to wild-type proviruses expressing Rem. The difference in mutations between Rem-expressing and non-expressing MMTV strains largely disappeared in mice lacking the Apobec family member, activation-induced cytidine deaminase (AID). These results suggested that Rem is an AID antagonist. In this study, we attempted to study AID-mediated mutations of TBLV proviruses lacking Rem expression obtained from tumors in C57BL/6 (B6) wild-type and AID-knockout backgrounds. Surprisingly, no differences in G-to-A mutations were observed in TBLV proviruses regardless of Rem expression, yet such mutations were significantly reduced in proviruses obtained from mA3/AID-double knockout mice relative to those from wild-type B6 or AID-knockout mice. Many cellular mRNAs involving the innate immune response, but not Apobecs, were elevated in the absence relative to the presence of Rem expression on the B6 AID-knockout background. These results revealed that Apobec-mediated mutagenesis is dependent on mouse strain and suggested a second means of Rem-dependent immune evasion.

## Introduction

Retroviruses are small RNA-containing viruses that replicate through a DNA intermediate to establish lifelong infections of their hosts [1]. To maintain long-term infections, retroviruses have developed various strategies to thwart the host immune response [2,3]. The human APO-BEC family cytidine deaminases contribute to innate immunity against RNA- and DNA-containing viruses, including retroviruses, as well as retrotransposons [2,4,5]. Retroviral replication often is inhibited by incorporation of these enzymes into virions [6–8] since human APOBEC3 (A3) packaging results in blocks to reverse transcription and mutagenesis of the proviral genome [9,10]. A3-mediated cytidine deamination on the human immunodeficiency virus type 1 (HIV-1) DNA negative strand in the absence of Vif leads to G-to-A sequence changes on the plus strand and inhibition of replication [11–13]. HIV-1-encoded Vif acts as an adapter to Cullin-based E3 ligases, which lead to ubiquitylation and proteasomal degradation of human A3G and 3F [14,15]. Nonetheless, many aspects of the immune response cannot feasibly be studied in humans. In contrast, mouse genetics has provided many important insights into the conserved biology of viruses and the antiviral immune response, including the role of Apobec proteins [16].

Experiments using the betaretrovirus mouse mammary tumor virus (MMTV) were the first to show inhibition of retroviral replication by mouse Apobec enzymes *in vivo* [6]. Unlike humans, mice have a single *Apobec3* (*mA3*) gene [2]. MMTV-RIII strain infection of C57BL/6 (B6) mice lacking a functional *mA3* gene showed accelerated viral replication and higher proviral loads compared to those in wild-type B6 mice. Interestingly, MMTV hypermutation was not observed in the presence of mA3 despite its packaging into MMTV virions [6]. Subsequent studies indicated that mA3 blocked MMTV reverse transcription [17], suggesting that mA3 does not induce hypermutation of the MMTV proviral genome.

Our studies in BALB/c mice showed that the Apobec family enzyme activation-induced cytidine deaminase (AID), which typically gives G-to-A or C-to-T mutations in immunoglobulin variable regions [18], leads to MMTV hypermutation in the absence of the virally encoded Rem protein [19]. MMTV proviruses lacking Rem expression (MMTV-SD) had increased

cytidine mutations within the AID-associated WR<u>C</u> (W = A/T, R = A/G) sequence motif, and also in TY<u>C</u> (Y = C/T) motifs associated with mA3 expression compared to wild-type MMTV proviruses (MMTV-WT). Both types of mutations were greatly reduced in MMTV-SD-induced tumors from mice lacking AID expression [19]. Transfection experiments showed that Rem co-expression leads to proteasomal degradation of AID, but not mA3. These experiments suggested that MMTV Rem is the functional equivalent of HIV-1 Vif and antagonizes AID, and possibly other Apobec family enzymes [19].

Rem is translated from a doubly spliced version of the envelope gene and in the same reading frame [20,21]. Both Env and Rem are translated at the endoplasmic reticulum (ER) membrane. Their common signal peptide (SP) is cleaved by signal peptidase [22–24] and retrotranslocated to yield a Rev-like protein that traffics to the nucleus for viral RNA export and expression [22,25,26] (Fig 1A). Both MMTV-WT and MMTV-SD produce SP from signal peptidase cleavage of the Env precursor. Thus, the Apobec-induced hypermutation phenotype of MMTV-SD is due to Rem C-terminal sequences, which include uncleaved Rem or the C-terminal cleavage product, Rem-CT [19]. Since Rem-CT is produced entirely within the ER and traffics within endosomal membranes, uncleaved Rem likely is the Apobec antagonist [27].

MMTV transmission to the mammary gland prior to tumor induction requires replication in both B and T cells [28–31], which are known to express AID and mA3 [32–34]. Apobec effects on retroviral replication *in vivo* primarily have been studied on the C57BL/6 (B6) background because of the ease of genetic manipulation [35], yet different mouse strains have distinct immune responses to pathogens [36]. To further understand the involvement of cytidine

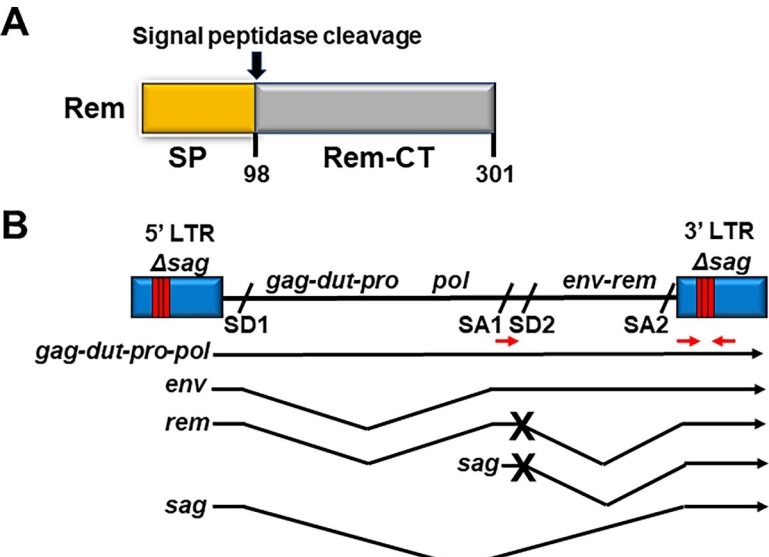

**Fig 1. Diagram of Rem and clonal infectious TBLV proviruses.** (A) Domain organization of Rem and its cleavage products SP and Rem-CT. The arrow indicates the position of Rem cleavage by signal peptidase. (B) Diagram of infectious TBLV-WT and TBLV-SD proviruses and their mRNA transcripts. The relative positions of genes are indicated on the proviral DNA. Both TBLV-WT and TBLV-SD have a T-cell enhancer that consists of a deletion and triplication of sequences flanking the deletion within the LTRs (red bars) compared to MMTV [37,52]. The LTR alteration leads to elimination of Sag expression, which is required for MMTV transmission and mammary tumorigenesis, but allows development of T-cell tumors [39]. The splice donor (SD) and acceptor (SA) sites are indicated on the provirus. TBLV-WT transcripts are shown by black arrows, and introns are designated by V shapes. Red arrows indicate primer positions for production of *env*-LTR fragments as well as shorter fragments containing LTR enhancer repeats. The TBLV-SD provirus has a mutation (marked by an "X") in SD site 2 (SD2), which prevents the synthesis of the doubly spliced *rem* mRNA [83] and *sag* transcripts from the intragenic promoter. Since TBLV does not have a functional *sag* open reading frame, TBLV-WT and TBLV-SD differ only by the production of *rem* mRNA.

deaminases in MMTV replication, we conducted additional experiments in B6 mice using the Sag-independent MMTV strain (TBLV-WT), which induces T-cell lymphomas rather than breast cancers [37–39] (Fig 1B). MMTV-encoded Sag protein on the surface of mature B cells is required for MMTV transmission from maternal milk in the gut to the mammary glands [40,41]. In this report, mutant TBLV lacking Rem expression (TBLV-SD) accelerated T-cell tumor induction compared to TBLV-WT in wild-type as well as AID-deficient $Aicda^{-/-}$ B6 mice [18]. Unlike the increased proviral mutations observed in the absence of Rem in BALB/c mice, TBLV-WT and TBLV-SD proviruses from B6 tumors had similar frequencies of mutations in the envelope region. Furthermore, RNA-seq analysis revealed increased expression of genes involved in growth factor and cytokine signaling in TBLV-SD-induced tumors relative to TBLV-WT-induced tumors, but only in AID-deficient, not wild-type B6 mice. The results suggest that the latency of TBLV-induced tumors in B6 mice is dependent on expression of the Rem C-terminus, but that proviral mutagenesis is dependent on mouse strain.

## Results

### Accelerated T-cell tumors induced by lymphomagenic MMTV (TBLV) in the absence of Rem and mature B cells

Our previous results using splice donor site mutants of MMTV and TBLV in BALB/c mice indicated that MMTV-encoded Rem is involved in antagonizing proviral mutagenesis by Apobec family enzymes, including AID [19]. Although Sag-mediated amplification of infected B and T lymphocytes is needed for efficient MMTV transmission and mammary cancer development [28,31,42], TBLV does not require Sag for the induction of T-cell lymphomas [39]. Therefore, using Sag-independent TBLV, we tested the effect of Rem loss in multiple strains of genetically engineered mice that are available on the B6 background. We inoculated TBLV-WT and TBLV-SD independently into B6 mice. As observed for TBLV-induced tumors in BALB/c mice [19], no statistically significant difference in tumor incidence or latency was observed in wild-type B6 mice inoculated with TBLV-WT or TBLV-SD (Fig 2A). However, most TBLV-WT-inoculated B6 mice developed T-cell lymphomas whereas, at the same dose, we observed a tumor incidence of 30–50% in BALB/c mice [19,39]. All TBLV-SD-injected B6 animals developed tumors, but there was a trend toward acceleration of T-cell tumors in the absence compared to the presence of Rem (p = 0.076).

Since our previous data suggest that Rem acts as a Vif-like antagonist of the Apobec family member AID in BALB/c mice and leads to AID proteasomal degradation in tissue culture [19], we also infected $Aicda^{-/-}$ mice on the B6 background with TBLV-WT or TBLV-SD. Like wild-type B6 mice, most animals developed T-cell lymphomas, and many had both spleen and lymph node involvement. However, no statistical difference in tumor incidence or latency was observed between mice inoculated with TBLV-WT relative to those inoculated with TBLV-SD (p = .909) (Fig 2B).

To determine if TBLV, like MMTV, requires replication in mature B cells, B6 µMT (lacking membrane-bound IgM) mice and wild-type B6 mice were infected with TBLV-WT or TBLV-SD. The µMT strain was developed by introducing a neomycin-resistance cassette to disrupt one of the membrane exons of the IgM heavy chain gene [43]. No expression of membrane-bound IgM is detectable and peripheral B-2 cells are lacking in this strain, although innate B-1 cells are made and produce IgE and IgG [44]. Increased plasmacytoid dendritic cells also are observed [45]. TBLV-WT induced tumors in µMT mice with ~100% efficiency (Fig 2C), and the latency was not significantly different from that in wild-type B6 mice (Fig 2A). Our results indicated that TBLV, a Sag-independent MMTV strain, does not require mature B-2 cells for T-cell lymphomagenesis. We also assessed whether inoculation of

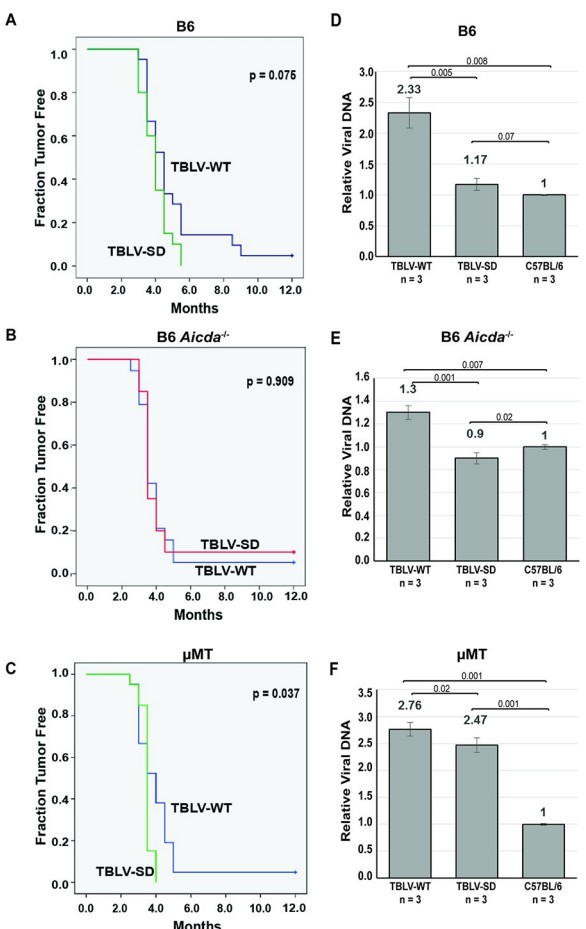

**Fig 2. T-cell tumor development after high-dose infection reveals that TBLV infection does not require mature B-2 cells.** (A-C) Wild-type B6, *Aicda*[-/-], or μMT mice (panels A to C, respectively) were injected with TBLV-WT or TBLV-SD (20–22 animals for each virus/strain combination) and followed for tumor development, including enlarged thymus, spleen, and lymph nodes. The results were analyzed by Kaplan-Meier plots. Only μMT mice showed a significant difference between T-cell lymphoma development by TBLV-WT and TBLV-SD at this viral inoculum (p<0.05) as determined by Mantel-Cox log-rank tests. TBLV-SD tumors were accelerated in μMT mice relative to the other two strains, but survival plots did not differ for TBLV-WT among the three strains. (D-F) Proviral loads were assessed in three tumors from each viral strain by semi-quantitative PCR, which allowed quantitation relative to the three diploid copies of endogenous *Mtv*s in B6 mice (panels D to F, respectively). Proviral load appeared to be lowest in *Aicda*[-/-] mice, yet proviral copies did not correlate with tumor latency. In some cases, acquired proviruses (haploid copies) could not be detected above the level of endogenous *Mtv*s (diploid copies). The relative proviral copy numbers appeared to be highest for TBLV-SD in μMT mice that lack mature B-2 cells.

TBLV-SD (lacking Rem expression) into μMT B6 mice would affect tumor incidence or latency compared to TBLV-WT. TBLV-SD infection led to a statistically accelerated appearance of T-cell lymphomas compared to those induced after TBLV-WT infection (p = 0.037) (Fig 2C). These data suggested that the loss of Rem expression by TBLV provided a selective advantage for virus replication and/or tumorigenesis in μMT mice.

## Decreased TBLV proviral loads in the absence of Rem depends on the presence of mature B cells

To determine if Rem expression affected proviral DNA levels, we extracted DNA from three independent TBLV-WT-induced tumors as well as the same number of TBLV-SD-induced

tumors for analysis by PCR. Because B6 mice contain three endogenous *Mtv* proviruses with extensive sequence similarity to TBLV [46,47], semi-quantitative PCRs that generated larger products allowed us to quantitate levels of haploid exogenous virus integrations relative to the diploid endogenous proviruses within tumor DNA [19]. TBLV-SD-induced tumors in B6 mice had a proviral load that was statistically lower compared to tumors induced by TBLV-WT (Fig 2D). These results suggested that the absence of Rem restricted TBLV replication prior to or during replication in T cells. Interestingly, proviral loads in *Aicda*$^{-/-}$ mice were lower in TBLV-WT-induced tumors (Fig 2E) compared to those in wild-type B6 mice (Fig 2D), consistent with decreased replication of Rem-expressing virus in the absence of AID expression. In contrast, proviral loads in TBLV-WT-induced tumors from μMT mice were similar to those induced in B6 mice (compare Fig 2D and 2F), and the proviral load was twice that observed in *Aicda*$^{-/-}$ mice (Fig 2E). Interestingly, the TBLV-SD proviral load in μMT tumors was ~2-fold increased over that observed in either wild-type or *Aicda*$^{-/-}$ mice.

## TBLV mutational profiles in the absence of Rem

Apobec family members are known restriction factors for retroviruses, including MMTV [6,19]. Proviral loads and transition mutations in WR$\underline{C}$ and TY$\underline{C}$ motifs typical of AID and mA3, respectively, were increased in BALB/c mammary tumors induced by MMTV lacking Rem expression. This difference was eliminated by induction of mammary tumors in AID-deficient *Aicda*$^{-/-}$ mice, suggesting that Rem protects against multiple Apobec cytidine deaminases [19]. In addition, previous high throughput sequencing of proviral DNA from BALB/c tumors showed a large increase in transition mutations in proviruses from TBLV-SD-induced tumors compared to those induced by TBLV-WT [19]. The frequency of C-to-T mutations on the proviral plus strand (average number of mutations/clone) showed a dramatic increase from 0.08 to 0.73 (~9-fold) when comparing TBLV-WT to TBLV-SD proviruses lacking Rem expression isolated from BALB/c tumors [19].

To assess whether proviral load differences between T-cell tumors induced by TBLV-WT and TBLV-SD were associated with alterations in Apobec-mediated hypermutation, tumor DNAs from B6 mice were amplified using TBLV-specific primers spanning the *env*-LTR region. Individual PCR products were cloned and analyzed by Sanger sequencing. In contrast to TBLV-induced BALB/c tumors [19], the average frequency of C-to-T mutations was similar for TBLV-WT and TBLV-SD proviruses (0.72 and 1.04 mutations/clone, respectively) obtained from B6 tumors (Table 1). These results suggested that MMTV-encoded Rem has little effect on Apobec-mediated proviral mutations in tumors induced in B6 relative to BALB/c mice.

Since the sequence context of Apobec-induced proviral mutations often reveals the identity of the enzyme involved [48], we analyzed mutations within specific sequence motifs. Analysis of mutations typical of murine AID (WR$\underline{C}$ motif) indicated a small increase for the TBLV-SD envelope region relative to TBLV-WT proviruses (1.5-fold) in B6 tumors (Table 1). In contrast, WR$\underline{C}$-motif (W = A or T, R = A or G) mutations were increased by 2.6-fold in TBLV-SD compared to TBLV-WT proviruses obtained from BALB/c tumors [19].

Proviral mutations typical of mA3 (TY$\underline{C}$; Y = T or C) were only 1.3-fold higher in TBLV-WT compared to TBLV-SD proviruses isolated from B6 tumors (Table 1), whereas mutations were 2-fold higher in TBLV-SD versus TBLV-WT proviruses from BALB/c tumors [19]. SY$\underline{C}$-motif (S = C or G, Y = T or C) mutations, which have been associated with AID enzymatic activity at lower frequency on immunoglobulin genes [49,50], were 4.9-fold higher for TBLV-SD relative to TBLV-WT proviruses in BALB/c mice [19]. However, this value was only 1.5-fold greater in TBLV-SD proviruses compared to TBLV-WT proviruses in B6 mice

**Table 1. Mutation frequency in TBLV-WT and TBLV-SD proviruses from B6 T-cell tumors by Sanger sequencing.**

| Mutation | TBLV-WT Mutation Frequency[1] | TBLV-SD Mutation Frequency[1] | Fold Increase (SD/WT) |
|---|---|---|---|
| G to A | 3.06 | 4.34 | 1.4 |
| A to G | 0.96 | 1.79 | 1.9 |
| C to T | 0.72 | 1.04 | 1.4 |
| T to C | 1.10 | 1.32 | 1.2 |
| Total Transitions | 5.84 | 8.49 | 1.5 |
| WRC | 0.42 | 0.64 | 1.5 |
| SYC | 0.36 | 0.53 | 1.5 |
| TYC | 2.26 | 2.89 | 1.3 |
| ATC | 0.22 | 1.10 | 5.0 |

[1] Mutations/number of clones. Based on Sanger sequencing of 1,100 bp of the plus strand of the proviral *env* gene clones obtained from three independent B6 tumors induced by TBLV-WT (n = 50) or TBLV-SD (n = 53).

(Table 1). The average frequency of ATC motif mutations, which have been associated with mA3 activity *in vitro* [51], was increased 2.6-fold for TBLV-SD relative to TBLV-WT proviruses in BALB/c mice (0.47 versus 0.18 mutations/clone, respectively) [19]. A 5-fold increase of ATC-motif mutations was observed in proviruses isolated from B6 mice (1.10 versus 0.22 mutations/clone for TBLV-SD and TBLV-WT, respectively) (Table 1). Therefore, the proviral mutations most affected by the absence of Rem in B6 tumors were those associated with the ATC motif, although the frequency of cytidine mutations remained highest in the TYC motif for either TBLV-WT or TBLV-SD. These data suggested that mutations in the ATC and TYC motifs are not produced by the same enzymes. Furthermore, greatly increased proviral mutations were observed in BALB/c tumors induced by TBLV-SD compared to TBLV-WT, but this difference was not detected in B6 tumors.

## Apobec-mediated mutations in TBLV proviruses related to mouse genetic background and Rem expression

Sanger sequencing of individual proviral clones from tumors induced in wild-type B6 as well as *Aicda*[-/-] mice was used to compare the distribution of cytidine mutations, which would be expected to occur on the negative retroviral strand. G-to-A mutations expected on the positive strand after reverse transcription were not increased in proviruses from *Aicda*-deficient mice infected with TBLV-SD relative to those in TBLV-WT-infected mice (Table 2). Scatter plot comparisons between tumors from wild-type mice revealed that only ATC-motif proviral mutations were increased after infection with TBLV lacking Rem expression. However, TYC-motif mutations were the most abundant in proviruses from both TBLV-WT and SD-infected mice (Fig 3A). Analysis of the distribution of ATC-motif mutations also revealed a significant increase within proviruses from tumors obtained from *Aicda*[-/-] mice infected with TBLV-SD relative to TBLV-WT (Fig 3B). Surprisingly, a significant increase was observed in WRC-motif mutations between TBLV-SD and TBLV-WT proviruses obtained from tumors lacking AID expression.

The distribution of specific cytidine mutations also was examined in TBLV proviruses obtained from µMT tumors. The G-to-A mutations on the TBLV-SD plus strand were 2-fold higher than those on the plus strand of TBLV-WT proviruses (Table 3). Unlike proviruses

**Table 2. Mutation frequency in TBLV-WT and TBLV-SD proviruses from B6 *Aicda*[-/-] T-cell tumors by Sanger sequencing.**

| Mutation | TBLV-WT Mutation Frequency[1] | TBLV-SD Mutation Frequency[1] | Fold Increase (SD/WT) |
|---|---|---|---|
| G to A | 2.38 | 2.09 | 0.9 |
| A to G | 0.58 | 0.60 | 1.0 |
| C to T | 0.22 | 0.35 | 1.6 |
| T to C | 0.78 | 0.65 | 0.8 |
| Total Transitions | 3.96 | 3.69 | 1.0 |
| WRC | 0.16 | 0.60 | 3.8 |
| SYC | 0.54 | 0.56 | 1.0 |
| TYC | 1.58 | 1.42 | 0.9 |
| ATC | 0.16 | 0.55 | 3.4 |

[1] Mutations/number of clones. Based on Sanger sequencing of 1,100 bp of the plus strand of the proviral *env* gene clones obtained from three independent B6 AID-KO tumors induced by TBLV-WT (n = 50) or TBLV-SD (n = 55).

from wild-type B6 and *Aicda*[-/-] mice, ATC-motif proviral mutations were similar in tumors induced by TBLV-WT and TBLV-SD (Fig 3C). TYC-motif proviral mutations were significantly increased in TBLV-SD-induced tumors relative to those induced by TBLV-WT.

The distributions of proviral TYC and ATC-motif mutations were dissimilar between tumors induced by TBLV-SD and TBLV-WT in all three backgrounds examined. For example, in wild-type B6 tumors, the TYC-motif mutations were not significantly different, whereas the ATC-motif mutations were significantly different. The opposite was observed in μMT mice. If mA3 was responsible for mutations in both ATC and TYC-motifs, the absence of Rem should have affected both types of mutations in the same way. Similarly, if WRC and SYC proviral mutations were both due to AID activity, we would anticipate that AID-knockout would have the same effect on each motif. Therefore, it appears likely that cytidine deaminases other than AID and mA3 cause the SYC and ATC-motif mutations.

## Lack of Rem expression accelerates wild-type B6 and AID-deficient tumors after low-dose infection

To determine whether differences between tumor susceptibility of B6 mice to TBLV-WT and TBLV-SD were dose-dependent, we injected a 20-fold lower dose into both wild-type B6 and *Aicda*[-/-] mice. At this dose, we achieved approximately the same tumor incidence in B6 mice injected with TBLV-WT as previously observed for BALB/c mice [19]. The results were plotted by the Kaplan-Meier method (Fig 4A). In contrast to the higher dose, TBLV-SD-induced tumor development was accelerated compared to that observed in TBLV-WT-injected mice in both mouse strains (p<0.0001). No statistical difference in tumor latency or incidence was observed in comparisons between TBLV-WT or TBLV-SD in the two strains. These data argue that Rem interferes with TBLV-induced tumor development independently of AID expression.

To determine whether the difference in tumor latency after TBLV-WT and SD infections was due to Apobec-mediated mutagenesis, clones spanning the *env*-LTR proviral region were obtained from three different B6 and *Aicda*[-/-] tumors and subjected to Sanger sequencing. Independent clones then were analyzed for cytidine mutations in specific sequence contexts within the *env* region. WRC-motif mutations were detectable at low levels, but their

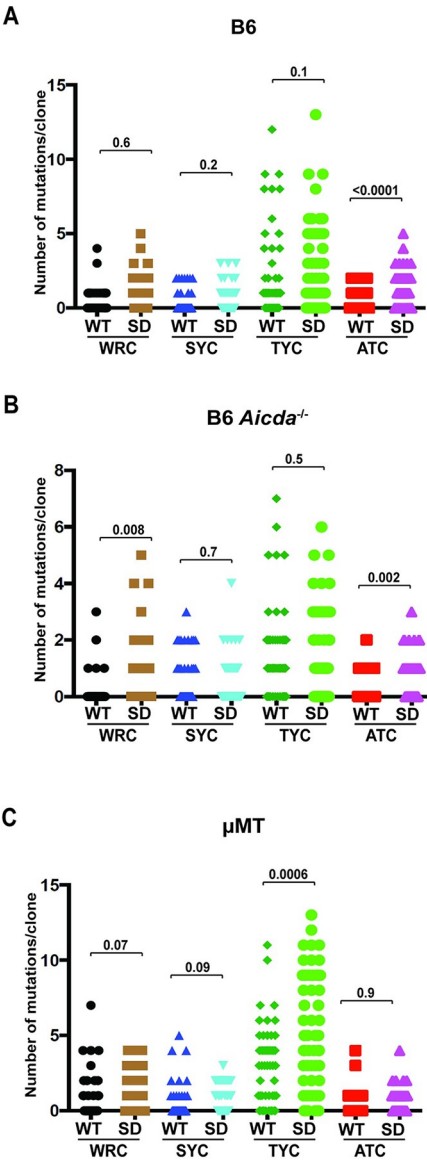

**Fig 3. Apobec-associated proviral mutations in tumors induced by high doses of TBLV-WT and TBLV-SD.**
Independent cloned sequences were obtained from three tumors in different animals. The number of cytidine mutations within different motifs on either proviral strand is given for each clone. WRC and SYC-motif mutations have been associated with AID expression, whereas TYC and ATC-motif mutations have been linked to mA3 expression. Statistical significance by non-parametric Mann-Whitney tests is indicated on the scatter plots. (A) Comparison of the distribution of mutations within the proviral envelope gene from TBLV-WT or TBLV-SD-induced tumors from wild-type B6 mice. (B) Comparison of the distribution of mutations within the proviral envelope gene in TBLV-WT or TBLV-SD-induced tumors from *Aicda*[-/-] mice. (C) Comparison of the distribution of mutations within the proviral envelope gene from TBLV-WT or TBLV-SD-induced tumors from μMT mice lacking mature B cells. Between 50–55 independent clones were analyzed for each virus/strain combination.

distributions were not different between either TBLV-WT or SD proviruses within or between wild-type and knockout mice (Fig 4C and 4D). In addition, the numbers of WRC-motif mutations/clone were similar in the presence or absence of a functional *Aicda* gene. Therefore, the cytidine mutations in WRC motifs are likely due to another deaminase.

**Table 3. Mutation frequency in TBLV-WT and TBLV-SD proviruses from B6 μMT T-cell tumors by Sanger sequencing.**

| Mutation | TBLV-WT Mutation Frequency[1] | TBLV-SD Mutation Frequency[1] | Fold Increase (SD/WT) |
|---|---|---|---|
| G to A | 3.74 | 7.59 | 2.0 |
| A to G | 0.74 | 0.57 | 0.8 |
| C to T | 0.54 | 0.43 | 0.8 |
| T to C | 0.98 | 0.45 | 0.5 |
| Total Transitions | 6.00 | 9.04 | 1.5 |
| WRC | 0.72 | 1.02 | 1.4 |
| SYC | 0.52 | 0.69 | 1.3 |
| TYC | 3.10 | 5.61 | 1.8 |
| ATC | 0.40 | 0.41 | 1.0 |

[1] Mutations/number of clones. Based on Sanger sequencing of 1,100 bp of the plus strand of the proviral *env* gene clones obtained from three independent B6 μMT tumors induced by TBLV-WT (n = 50) or TBLV-SD (n = 51).

As noted in high-dose tumors, tumors derived from lower-dose infections had the most abundant proviral mutations in the TYC context typical of mA3. Similar to cytidine mutations in the WRC context, no statistical differences in the distribution of TYC-motif mutations were detected between wild-type and Rem-null proviruses in either B6 or *Aicda*[-/-] mice (Fig 4C and 4D). Also, unlike our results with MMTV-induced tumors in BALB/c mice [19], the numbers of TYC-motif mutations/clone did not decline in the absence of AID expression. No significant differences were observed in the SYC or ATC contexts in the presence or absence of Rem or AID expression. These results suggest that Apobec-mediated hypermutations are not responsible for tumor latency differences observed between TBLV-WT and TBLV-SD infections and are mouse-strain-dependent.

To increase the number of proviruses analyzed for Apobec-mediated changes, we performed MiSeq sequencing on the PCR products spanning the *env*-LTR region of proviruses derived from tumors appearing after the high-dose TBLV infection (Fig 5A). As expected, analysis revealed that G-to-A mutations typical of Apobecs were the most abundant changes, but no significant differences were observed between TBLV-WT and SD proviruses in either wild-type or *Aicda*-deficient B6 mice. We also analyzed tumors from low-dose infections by Nanopore sequencing of integrated proviruses (Fig 5B). In this experiment, we included tumors induced by TBLV-WT or SD infection of wild-type, *Aicda*-deficient, and *Aicda/mA3*-deficient B6 mice. Consistent with the results of MiSeq data, no statistical differences in proviral mutagenesis were detected between wild-type and AID-knockout mice. In contrast, both TBLV-WT and SD proviruses had significantly lower G-to-A mutations when mA3 as well as AID expression was eliminated. Our data suggest that mA3 is responsible for a large number of G-to-A proviral mutations, although the levels in proviruses from double-knockout mice were still higher than for other types of sequence alterations. These results further confirm that Rem expression had no demonstrable effect on TBLV proviral mutations in tumors induced in B6 mice.

To test whether TBLV-induced tumors from wild-type or *Aicda*-deficient mice had altered levels of mA3, we performed Western blotting on extracts from tumors of each genotype and infecting virus (S1A Fig). Levels of mA3 were quantified relative to Gapdh levels in the same extracts, and the means and standard deviations were determined for four independent tumors (S1B Fig). No significant differences were observed between mA3 levels regardless of

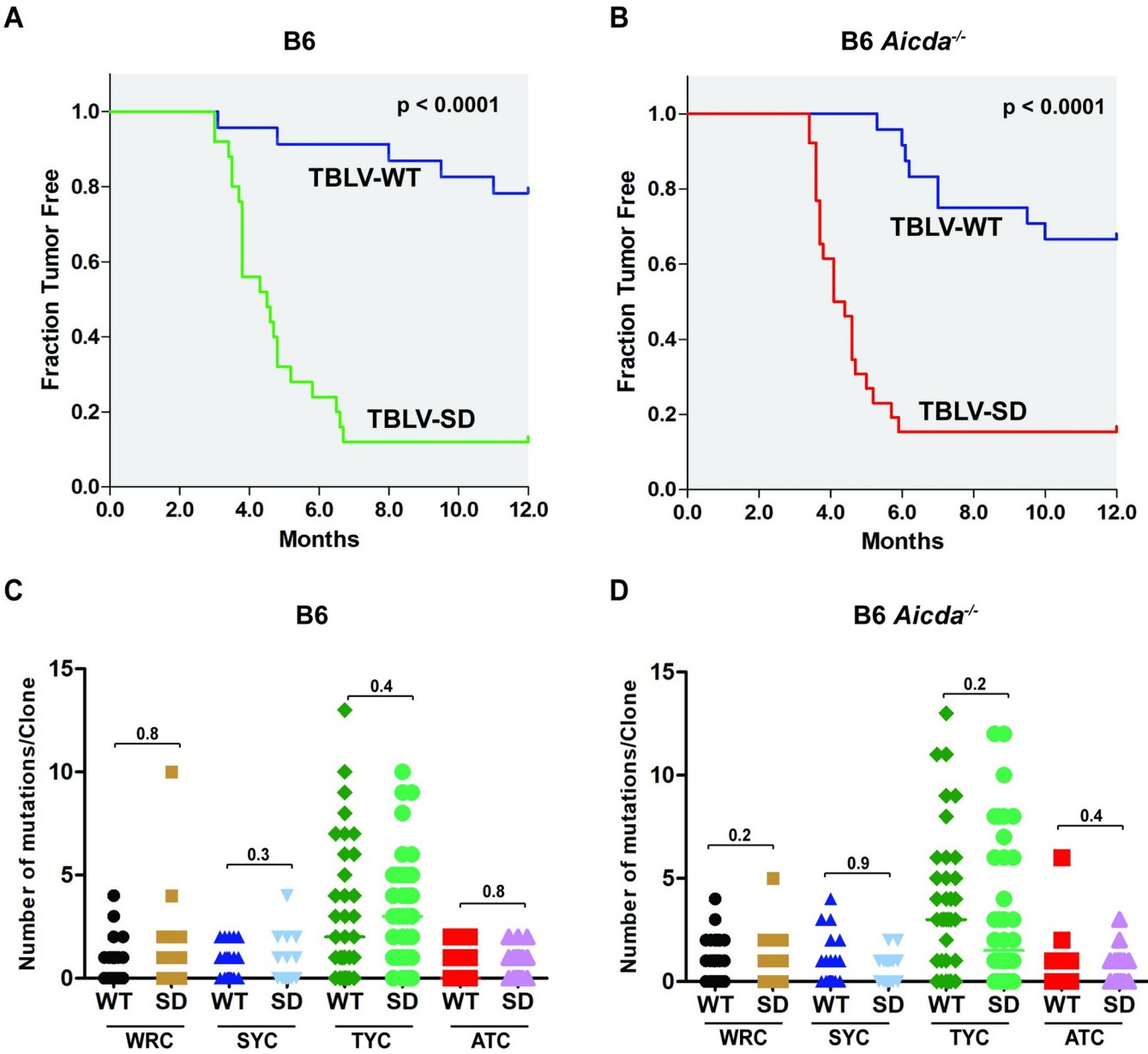

**Fig 4. Low-dose infections in either wild-type or *Aicda*⁻/⁻ B6 mice reveal accelerated tumorigenesis by TBLV in the absence of Rem expression.** (A) Kaplan-Meier plots of tumors induced by the same low dose of TBLV-WT or TBLV-SD in wild-type mice on the B6 background. This dose was 20-fold lower than that used for results shown in Fig 2. (B) Kaplan-Meier plots of tumors induced by the same low dose of TBLV-WT or TBLV-SD in *Aicda*⁻/⁻ B6 mice. The p-values were calculated by Mantel-Cox log-rank tests. (C) Sanger sequencing of individual TBLV proviral clones from low-dose tumors in wild-type B6 mice. Differences in the distribution of cytidine mutations within the envelope gene of cloned proviruses (~25 clones from three different tumors) were compared for WRC, SYC, TYC, and ATC motifs using scatter plots. The p-values were calculated by non-parametric Mann-Whitney tests. The most prevalent mutations were in the TYC motif typical of mA3. (D) Sanger sequencing of individual TBLV proviral clones from low-dose tumors in *Aicda*⁻/⁻ B6 mice. Comparisons were performed as described in panel C.

AID or Rem expression. As expected, AID expression was not detected in any of these T-cell tumors (Table 4). These data suggested that neither Rem nor AID influences mA3 levels in TBLV-induced T-cell tumors.

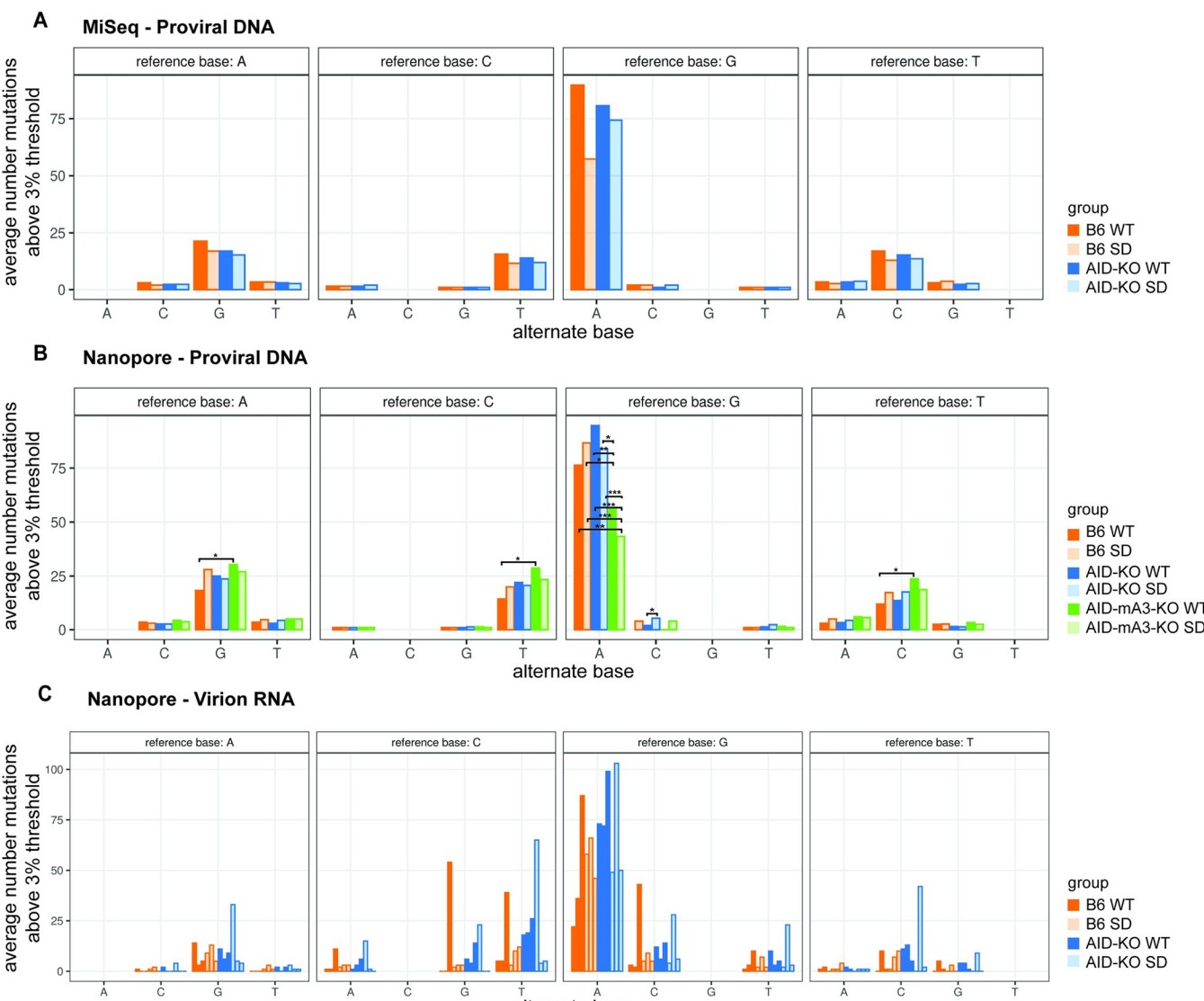

**Fig 5. High-throughput sequencing of proviruses from B6 tumors induced by infections of TBLV-WT and TBLV-SD.** (A) MiSeq sequencing of the *env*-LTR proviral junction fragments obtained by PCR using DNA from TBLV-WT or TBLV-SD-induced thymic lymphomas of infected wild-type (dark orange and light orange, respectively) or AID (*Aicda*)-knockout mice (dark blue and light blue, respectively). Tumors were obtained after high-dose injections. Each bar represents results from three independent tumors. The average number of mutations above a 3% threshold are plotted for each position relative to the cloned TBLV sequence. Pairwise comparisons between groups showed no statistical differences. (B) Nanopore sequencing of the *env*-LTR proviral junction fragments obtained by PCR using DNA from the *env*-LTR proviral junction fragments obtained by PCR using DNA from TBLV-WT or TBLV-SD-induced thymic lymphomas of infected wild-type (dark orange and light orange, respectively), AID-knockout mice (dark blue and light blue, respectively) or mA3-AID (*Aicda*) double-knockout mice (dark green and light green, respectively). Each bar represents results from three independent tumors obtained after low-dose injections. Statistical differences between virus/strain combinations are shown as *p<0.05, **p<0.01, or ***p<0.001. (C) Nanopore sequencing of the PCR products derived from virion RNA extracted from low-dose TBLV-WT or SD-induced thymic lymphomas. The numbers of mutations in virion RNAs extracted from each tumor were given separately due to the wide variation in the values observed. The color scheme is the same as in panel A. The mutational variability observed among the triplicate samples prevented statistical analysis.

## Virion RNA mutations in the absence of AID and Rem expression

The presence of significant G-to-A mutations in TBLV proviral DNA obtained from tumors prompted us to test whether only selected TBLV RNAs are packaged into virions in the absence of Rem expression. Three independent TBLV-WT or SD-induced tumors from B6 or

**Table 4. Correlation of Proviral Mutations with B6 and BALB/c Characteristics.**

| Characteristic | BALB/c | C57BL/6 |
|---|---|---|
| Immune bias | Th2 | Th1 |
| Increased proviral mutations without Rem | Yes | No |
| mA3 isoforms | exon5+>Δexon5 | Δexon5 only |
| Relative mA3 protein | Low | High (X-MuLV LTR+) |
| Response to ConA[1] | <2-fold mA3 increase; small AID increase | ~2-fold mA3 increase; no AID detected |
| Response to IL-4/LPS | ~2-5-fold mA3 increase; large AID increase | ~2-3-fold mA3 increase; large AID increase |

[1] Fold induction relative to untreated splenocytes.

*Aicda*[-/-] mice were used for isolation of virions, which were used for RNA extraction. The yield of viral RNA was very similar regardless of Rem or AID expression. RNA samples were subjected to reverse transcription-PCR (RT-PCR) with the same primers used for integrated TBLV proviruses to yield an ~2 kb envelope-LTR (*env*-LTR) junction fragment. The RT-PCR products were subjected to Nanopore sequencing.

To test whether the loss of *rem* and/or the *Aicda* genes led to selective virion incorporation of specific mutant viral RNAs, sequences were analyzed for mutations typical of cytidine deaminases (Fig 5C). G-to-A mutations were predominant, yet the numbers of mutations in virion RNA isolated from different tumors were quite variable. Although these data were not useful for statistical analysis, the results suggested that neither Rem nor AID affected the quantity or mutational profile of virion RNA, consistent with the proviral DNA analysis.

## Analysis of LTR enhancer repeats within tumor-derived TBLV proviruses

Acceleration of T-cell lymphomas was observed after infection with TBLV-SD compared to TBLV-WT infections in both wild-type and *Aicda*[-/-] B6 mice. We considered whether differential selection for *cis*-acting LTR elements, which are known to affect TBLV disease specificity [38,52,53], could accelerate TBLV-SD induction of tumors in μMT mice after Apobec-mediated mutagenesis. Specifically, TBLV proviruses have a triplicated T-cell enhancer element within the LTR (Fig 1) that increases viral transcription in T cells [37,52]. Our previous work has shown that changes within this region are selected during tumor passage, with 1- and 4-enhancer elements being less active transcriptionally than 3-enhancer repeats [53]. As expected, PCR performed with LTR-specific primers showed a predominance of 3 repeats within the LTR, consistent with the injected clonal virus [39]. No dramatic change in the number of enhancer repeats within TBLV-SD proviruses relative to TBLV-WT proviruses was observed in tumors from wild-type or AID-deficient (*Aicda*[-/-]) B6 mice that would confer an obvious selective advantage (Fig 6A and 6B). However, the results revealed that the viral populations within tumors were mixed, consistent with LTR selection during lymphomagenesis [53].

## Differential mA3 and AID levels and signaling in B6 and BALB/c mice

To understand the correlation between AID and mA3 expression with proviral hypermutation in BALB/c and B6 tumors, we isolated splenocytes from each strain. Pooled cells from multiple mice then were treated for increasing times between 4 and 96 h with interleukin-4 (IL-4) and lipopolysaccharide (LPS) to stimulate AID production in B cells. After incubation with these stimulants, cell lysates were used for Western blotting to determine both AID and mA3 levels.

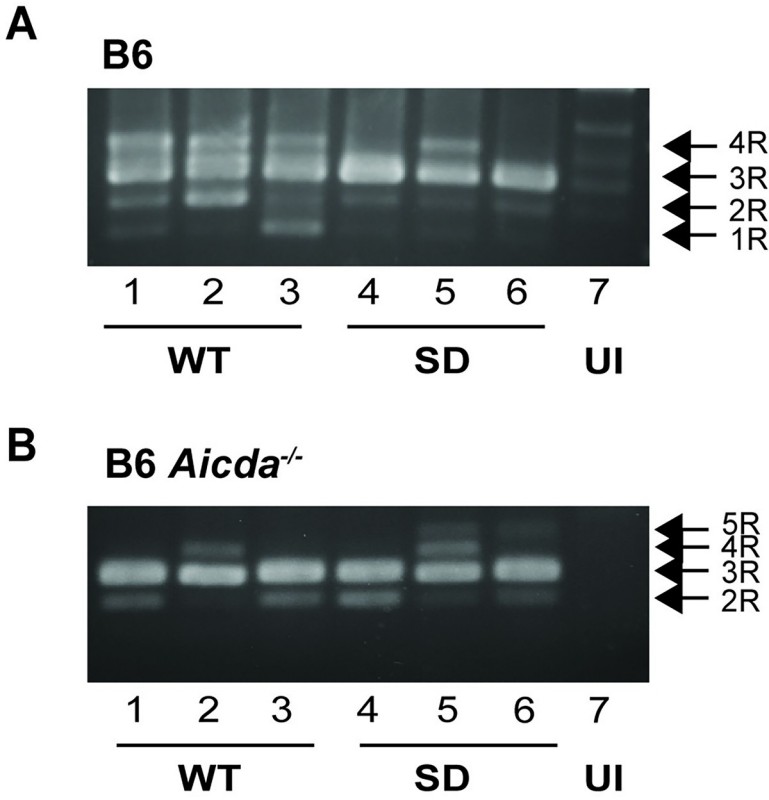

**Fig 6. LTR enhancer repeats within TBLV proviruses obtained from low-dose tumors induced in wild-type and** *Aicda*^-/-^ **mice.** (A) Analysis of LTR enhancer repeats in tumors induced by TBLV-WT (WT) and TBLV-SD (SD) in wild-type B6 mice. PCRs were performed on three tumors from each viral strain prior to agarose gel electrophoresis. The positions of LTRs containing 1 (1R), 2 (2R), 3 (3R), 4 (4R), or 5 (5R) copies of the 62-bp sequence in the TBLV enhancer [53] are indicated. The infectious TBLV-WT and TBLV-SD clones both have a 3R enhancer [39]. (B) Analysis of LTR enhancer repeats in tumors induced by TBLV-WT and TBLV-SD in *Aicda*^-/-^ mice on the B6 background. UI = DNA from uninfected mice. Faint background bands are observed in DNA from uninfected mice, presumably due to homology of the primers to the endogenous *Mtv*s.

As expected, constitutive mA3 levels were higher in B6 splenocytes relative to those obtained from BALB/c mice (Fig 7A, compare lanes 1 and 6), and only the Δexon5 isoform was detected [51,54].

Comparisons to extracts from uncultured splenocytes revealed that culture in the presence of IL-4/LPS gave a large increase in AID levels in BALB/c cells at 48 h. In contrast, mA3 levels showed a more modest increase in response to IL-4/LPS, yet the increased mA3 protein was detected within 4 h after stimulation. The mA3 levels were quantitated relative to levels in uncultured splenocytes in three different experiments (Fig 7B). Considerable variability in mA3 induction by IL-4/LPS was observed, particularly in B6 splenocytes, although lysates were derived from pooled splenocytes from 5 mice in each experiment. The results confirmed higher (~6-fold average) baseline mA3 expression in B6 relative to BALB/c splenocytes. IL-4/ LPS treatment gave ~2-5-fold and ~2-3-fold induction in BALB/c and B6 splenocytes, respectively, relative to baseline mA3 levels.

Concanavalin A (ConA) treatment gave little change in mA3 basal levels in BALB/c splenocytes, but had a larger effect in B6 cells (Fig 7C). The mA3 levels detected in three independent experiments were quantified relative to Gapdh and used to calculate means and standard deviations (Fig 7D). The mean increase in ConA-induced mA3 levels was ~1.5-fold over basal

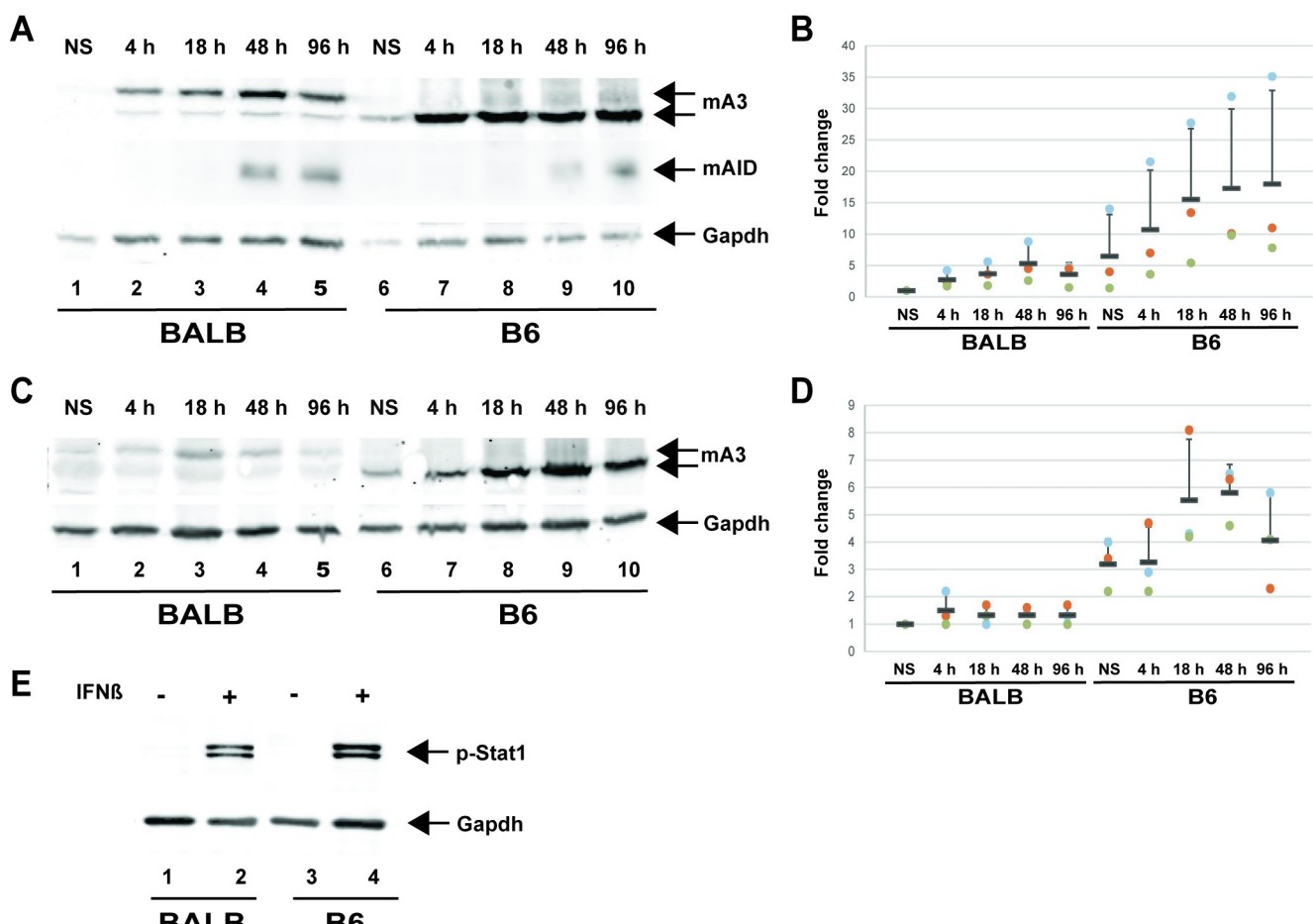

**Fig 7. Expression of mAID and mA3 differs in B6 and BALB/c splenocytes after external ligand stimulation.** (A) Splenocyte responses to IL-4 plus LPS. Splenocytes from uninfected BALB/c or B6 mice were used for whole cell lysate preparation prior to culture with IL-4 plus LPS for 4 h (lanes 2 and 7), 18 h (lanes 3 and 8), 48 h (lanes 4 and 9), or 96 h (lanes 5 and 10). Uncultured splenocyte lysates (NS = non-stimulated) are shown in lanes 1 and 6. Lysates were used for Western blotting and incubation with antibodies specific for mA3 (top), murine AID (mAID) (middle) or Gapdh (bottom). Note that BALB/c mice express two isoforms of mA3 mRNA (with and without exon 5), whereas B6 express predominantly the isoform without exon 5 [54,72]. (B) Quantitation of mA3 levels after stimulation with IL-4 and LPS in three independent experiments. Each experiment (shown with dots of different colors) represents pooled splenocytes from 5 animals. Means are shown by horizontal lines, whereas standard deviations are shown by vertical bars in one direction. (C) Splenocyte responses to Concanavalin A (ConA). Splenocytes from uninfected BALB/c or B6 mice were used for whole cell lysate preparation prior to culture with ConA for 4 h (lanes 2 and 7), 18 h (lanes 3 and 8), 48 h (lanes 4 and 9), or 96 h (lanes 5 and 10). Uncultured splenocyte lysates are shown in lanes 1 and 6. Lysates were used for Western blotting and incubation with antibodies specific for mA3 (top) or Gapdh (bottom). (D) Quantitation of mA3 levels after stimulation with ConA in three independent experiments. Each experiment (shown with dots of different colors) represents pooled splenocytes from 5 animals. Means are shown by horizontal lines, whereas standard deviations are shown by vertical bars in one direction. (E) Treatment with IFNβ gives similar signaling responses in BALB/c and B6 splenocytes. Lysates from splenocytes treated with or without 1000 U/ml IFNβ for 5 h were used for Western blotting with antibodies specific for phosphorylated Stat1 (p-Stat1) or Gapdh. Two bands observed with p-Stat1 antibody correspond to phosphorylated Stat1α and Stat1β isoforms and are expressed similarly in BALB/c and B6 mice.

BALB/c levels. In B6 splenocytes, the mean increase over basal mA3 levels was ~2-fold after 18 h in the presence of ConA. AID levels were slightly increased or not detectable in response to ConA stimulation of BALB/c or B6 splenocytes (Table 4). IFNβ treatment of BALB/c and B6 splenocyte populations revealed similar Stat1 signaling as demonstrated by phosphorylated Stat1 levels (Fig 7E). Together, these results indicated that there are differences in signaling pathways for AID induction in BALB/c and B6 lymphocytes as well as differences in constitutive and inducible mA3 and AID levels (see Table 4) [51,54].

### Effects of Rem and AID on chemokine and cytokine signaling

To determine differences in transcription that alter cell signaling induced in the presence and absence of Rem expression, we extracted RNA from three independent tumors induced by TBLV-WT or TBLV-SD after low-dose infection. In tumors from infected wild-type B6 mice, >15,000 transcripts were detected by RNA-seq. Using a 2-fold difference in mRNA levels as a cutoff, 30 transcripts were significantly upregulated in TBLV-SD-induced tumors relative to those induced by TBLV-WT. Nine transcripts, including *Ighg3* and *C4b*, were significantly downregulated (Fig 8A). Analysis of tumors from *Aicda*$^{-/-}$ mice infected with either TBLV-WT or TBLV-SD revealed a similar number of total mRNAs as those obtained from infected B6 tumors. In contrast to results with wild-type B6 mice, 424 different gene transcripts were significantly increased by at least 2-fold in AID-deficient tumors infected with TBLV-SD relative to those infected with TBLV-WT (Fig 8B). Three of the most highly upregulated transcripts in the absence of Rem expression were *Il2ra*, *Socs3*, and *Stat4*. In addition, 83 mRNAs were significantly downregulated in TBLV-SD-induced tumors when compared to mRNAs in TBLV-WT-induced tumors. The complete list of transcripts and their relative abundance after TBLV infection in B6 and *Aicda*$^{-/-}$ mice is provided in S1 and S2 Tables, respectively. Therefore, the most dramatic effect on the transcriptome of TBLV-induced tumors occurred in the absence of both AID and Rem expression.

To determine whether the genes upregulated in the absence of Rem and AID were associated with a particular cellular pathway, we used PANTHER analysis (www.pantherdb.org). The results revealed that >20 upregulated genes were associated with platelet-derived growth factor (PDGF) and chemokine/cytokine signaling (Fig 8C). Although Rem functions both in Apobec antagonism and control of viral mRNA nuclear export and expression [19–21], these activities targeted post-transcriptional functions, not mRNA levels.

To validate the results of the RNA-seq analysis, we selected two genes, *Socs3* and *Stat1*, in the *Aicda*$^{-/-}$ tumors induced by TBLV-SD for further testing. Four tumors each from TBLV-WT and TBLV-SD-infected B6 and *Aicda*$^{-/-}$ mice were used for reverse-transcription real-time PCR (Fig 8D). As expected, both *Socs3* and *Stat1* mRNAs were increased relative to glyceraldehyde-3-phosphate dehydrogenase (*Gapdh*) levels in TBLV-SD-infected *Aicda*-deficient tumors relative to those in tumors from TBLV-WT-infected animals. However, neither *Socs3* nor *Stat1* mRNAs were statistically elevated in tumors induced by TBLV-SD relative to those induced by TBLV-WT in wild-type B6 mice. These results suggest an interplay between Rem and AID activities that affects RNA levels of multiple genes, including those involving inflammation mediated by chemokine and cytokine signaling.

## Discussion

We previously showed that infectious MMTV and TBLV proviruses lacking Rem expression obtained from tumors on the BALB/c background have multiple transition mutations typical of Apobec cytidine deaminases [19]. The mutational differences between Rem-expressing and Rem-null proviruses were primarily in cytidine-containing motifs typical of AID and mA3 and were abolished in *Aicda*$^{-/-}$ BALB/c mice [19]. However, most mouse strains with mutations or deletions in one or more *Apobec* genes are on a B6 background, and previous experiments have shown that MMTV has higher viral loads in B6 *mA3*$^{-/-}$ compared to wild-type mice [6]. Therefore, we conducted experiments in B6 and *Apobec*-mutant mice to confirm the mutational phenotype observed in BALB/c mice. Surprisingly, only small mutational differences were observed between TBLV-SD and TBLV-WT proviruses from B6 tumors.

Because B6 mouse cells are unable to efficiently present C3H MMTV Sag due to lack of MHC class II I-E expression [55], the Sag-independent MMTV strain, TBLV [37,39], was used

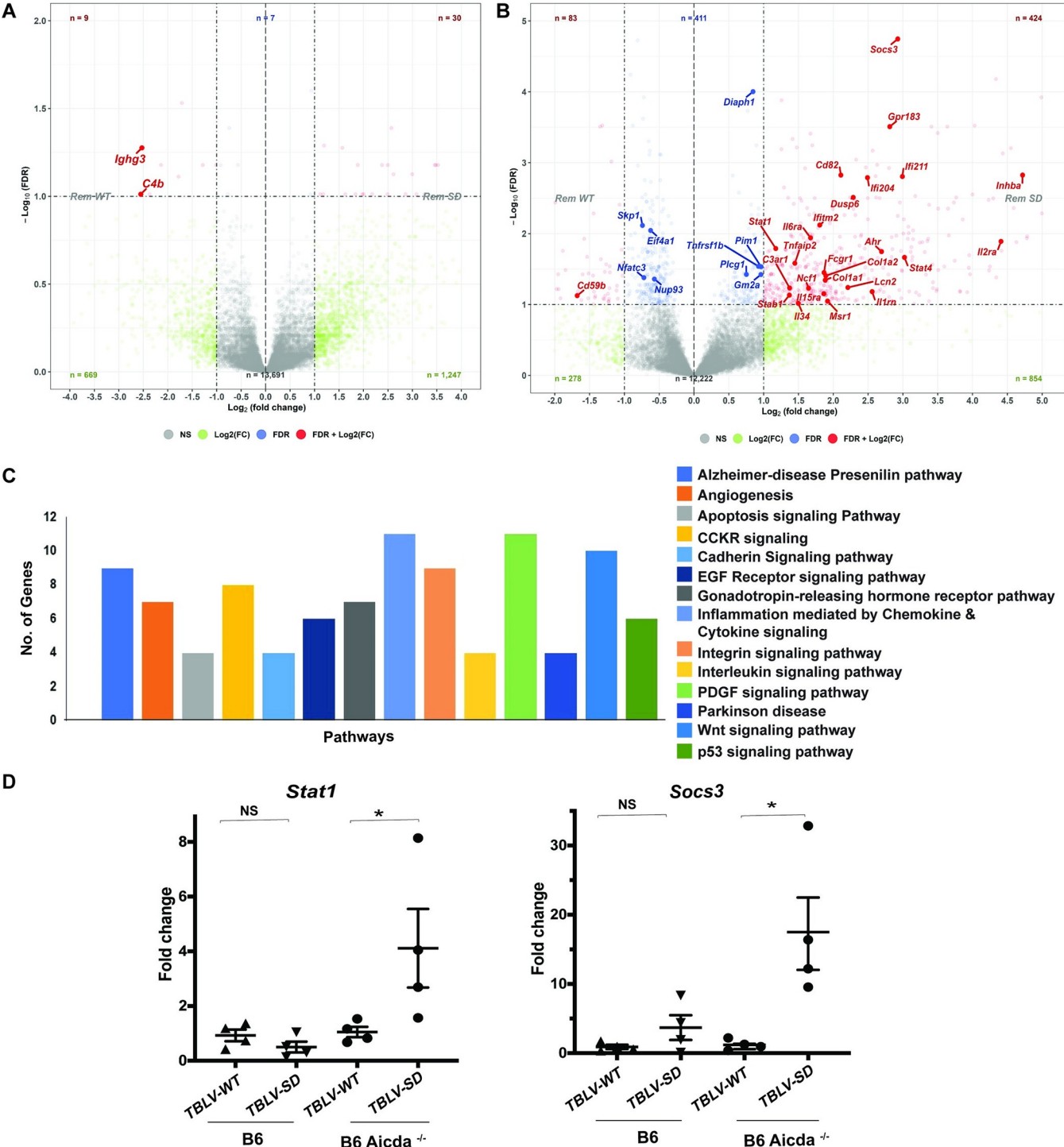

**Fig 8. RNA-seq analysis indicates increased immune-related transcripts in the absence of Rem and AID expression.** (A) Volcano plot comparing different transcripts from TBLV-WT and TBLV-SD-infected tumors from B6 mice. The false discovery rate (FDR) (-log₁₀) was plotted versus the log₂-fold change (FC) in mRNA abundance. Blue dots indicate transcripts that were significantly different in TBLV-SD-induced tumors. Red dots indicate transcripts that were significantly different in TBLV-SD-induced tumors and were changed more than 2-fold. Non-significant differences are shown by gray dots. The identities of some transcripts are provided. (B) Volcano plot comparing different transcripts from TBLV-WT and TBLV-SD-infected tumors from B6 *Aicda⁻/⁻* mice. (C) PANTHER analysis of differentially expressed genes in tumors induced by TBLV-SD relative to TBLV-WT. (D) Validation of *Stat1* and *Socs3* mRNAs expressed in wild-type and AID-deficient B6 mice in tumors induced by TBLV-WT and TBLV-SD (4 tumors each). Each symbol represents RNA from a single tumor. Mean and range of values obtained by RT-qPCR are shown. NS = non-significant.

for infection of B6 mice. BALB/c and B6 mouse strains are immunologically distinct [56]. In response to pathogens, T cells from B6 mice preferentially produce Th1 cytokines with high interferon-gamma (IFNγ) and low interleukin (IL-4), whereas those from BALB/c produce Th2 cytokines with low IFNγ and high IL-4 production [36]. BALB/c mice also have only a transient antibody response to MMTV infection [57]. Therefore, we anticipated that B6 mice would be more resistant to TBLV-induced tumors. Surprisingly, the TBLV-WT dose that gave a 30–50% tumor incidence in BALB/c mice [19,39] resulted in a nearly 100% incidence in B6 mice (Fig 2). In contrast, a 20-fold lower TBLV-WT dose produced approximately the same tumor incidence in BALB/c and B6 mice (Fig 4).

We also analyzed TBLV-induced tumors in B6 mice lacking mature B cells (μMT) [43] or AID expression (*Aicda*$^{-/-}$) [18]. At the higher dose, TBLV-SD-infected μMT mice had a significantly shorter tumor latency compared to TBLV-WT-infected animals, whereas AID-knockout mice showed no significant difference. However, at the lower dose, both AID-knockout and wild-type B6 mice had increased tumor incidence and decreased latency after infection with TBLV-SD relative to TBLV-WT. These results suggest that Rem expression affects tumor formation or progression.

Only a few insertions activate specific proto-oncogenes that provide a cell growth advantage [58–61]. Examination of proviral loads revealed that tumors in B6 mice had lower TBLV proviral loads in the absence of Rem expression. AID-knockout mice showed lower proviral loads between TBLV-WT and SD-induced tumors, yet overall loads were reduced compared to those of either B6 wild-type or μMT mice (Fig 2). These results suggest that mature B-2 cells, which are not required for TBLV transmission, inhibit virus replication. Decreased proviral loads in TBLV-WT-induced tumors from *Aicda*$^{-/-}$ mice compared to those from B6 or μMT tumors may indicate that AID-expressing cells are needed for optimal replication of a Rem+ virus.

Surprisingly, we observed much smaller differences in the numbers of mutations/clone between TBLV-WT and TBLV-SD proviruses obtained from B6 tumors (Figs 3–5) relative to those from BALB/c tumors [19]. In BALB/c tumors, TBLV-SD proviruses had increased TY<u>C</u> and WR<u>C</u>-motif mutations typical of mA3 and AID, respectively, relative to TBLV-WT proviruses [19]. These mutational differences between TBLV-WT and TBLV-SD proviruses largely were abolished in the absence of AID expression [19]. However, increased mutations/clone and significantly different distributions only were observed in mA3-associated TY<u>C</u> motifs in TBLV-SD versus TBLV-WT proviruses in μMT B6 mice, which lack mature B-2 cells [43], and not in either wild-type B6 or *Aicda*$^{-/-}$ mice (Fig 3). Previous results have shown that μMT mice have normal bone marrow levels of pro-B (B220$^{low}$, c-kit+, CD19-) and pre-B cells (B220$^{low}$, c-kit+, CD19+), but decreased levels of immature (B220$^{low}$, IgM+) and plasma (CD138+, TACI+) cells as well as greatly reduced spleen weight and total B cells (B220+) [62]. In contrast, significant differences in the distribution of AT<u>C</u>-motif mutations, a minor proportion of the total, within TBLV-WT and SD proviruses only were observed in wild-type B6 and AID-knockout tumors. This difference in proviral mutations was not observed in μMT tumors, perhaps due to the paucity of B-2 cells [45]. Although AT<u>C</u>-motif mutations have been attributed to mA3 from *in vitro* experiments [51], the differences in their distributions in the absence of Rem and in different mouse strains suggest that cytidine mutations within AT<u>C</u> and TY<u>C</u> sequences result from independent enzymes.

WR<u>C</u> and SY<u>C</u>-motif mutations both have been associated with AID-induced mutations of the immunoglobulin variable region genes, with mutations in WR<u>C</u> motifs being much more common [63]. Although AID is known to be responsible for antibody affinity maturation in germinal center B cells, AID-catalyzed somatic hypermutation and class-switch recombination also have been demonstrated in human immature B cells [64]. We observed significant

differences in the distribution of WRC-motif mutations between TBLV-WT and TBLV-SD-induced BALB/c tumors [19]. This difference was not found in wild-type B6 tumors using Sanger sequencing of individual clones within the envelope region (Fig 3). Unexpectedly, proviral mutations in WRC motifs were observed with a similar frequency and distribution after infection with TBLV-WT or TBLV-SD virus in wild-type B6 and μMT mice. However, in *Aicda*[-/-] mice, WRC-motif mutations increased in the absence of Rem expression. SYC-motif mutations did not follow this pattern, consistent with the idea that different enzymes induced cytidine mutations in the WRC and SYC context (Fig 3). On the other hand, Nanopore sequencing of proviruses recovered from AID-mA3-double knockout mice revealed a significant decline in G-to-A mutations compared to those from AID-knockout mice. Our data strongly suggest that mA3 mutates TBLV proviruses on the B6 background, which is consistent with the high levels of mA3 expression in B6 mice [54]. This result is in contrast to sequencing of RIII-strain MMTV proviruses after infection of B6 mice [6]. The remaining G-to-A mutations in proviruses obtained from the double AID-mA3-knockout mice may be due to additional cytidine deaminases that are not antagonized by MMTV-encoded Rem and require further investigation.

Replication of certain murine leukemia viruses (MuLVs) (gammaretroviruses) is inhibited by mA3 [54,65–69]. B6 infections with a glyco-Gag mutant of Moloney MuLV (MoMuLV) reduced infectivity, but this was abolished in *mA3*[-/-] mice [68]. Hypermutations were not observed in the absence of glyco-Gag in infected wild-type B6 mice. Infection of BALB/c mice with a glyco-Gag defective mutant also showed decreased titers relative to wild-type mice, yet proviral hypermutations were not examined [68]. Glyco-Gag is an N-terminal extension of the Gag structural capsid protein precursor that allows membrane insertion [70]. MuLVs without glyco-Gag expression appear to have a late-stage defect that involves budding and release in cultured fibroblasts [71]. Glyco-Gag is incorporated into virions to stabilize the capsid and prevents mA3 access to reverse transcripts [68]. Inhibition of mA3 by glycosylated Gag does not require deaminase function [69]. Nevertheless, certain MuLV strains, such as AKV, show hypermutation of the proviral genome even when glyco-Gag is expressed [65–67].

Co-transfection of mA3 from BALB/c (either exon5-plus or minus isoforms), NIH Swiss, or B6 mice into 293T cells together with AKV or MoMLV infectious clones revealed that AKV, not MoMuLV, proviruses were hypermutated after infection of NIH3T3 cells. The AKV hypermutations primarily occurred in the TTC motif [65], which is consistent with our use of the TYC motif. AKV restriction correlated with mA3 levels rather than specific mRNA isoforms, with B6 neonatal splenocytes having the highest amounts of mA3 protein relative to BALB/c or NIH Swiss splenocytes. Both rat A3 and human A3G hypermutated MoMuLV. Purification of splenic B and T cells from B6 mice and infection with MoMuLV or AKV clones revealed mA3-induced mutations within both cell types, but only in AKV proviruses [65]. These experiments suggested that the MuLV strain determined their susceptibility to mA3 hypermutation. In contrast, our results showed that the same viral strain (TBLV) displayed proviral mutations dependent on the mouse strain infected.

Low splenocyte levels of mA3 relative to AID correlated with tumor-associated proviral hypermutations in BALB/c mice, whereas the opposite was true in B6 mice (Fig 7). Moreover, BALB/c and B6 splenocytes responded differently to ConA and IL-4/LPS (summarized in Table 4). For example, ConA induced mA3 in both BALB/c and B6 splenocytes, whereas AID was induced to higher levels in BALB/c spleen cells (Fig 7). The difference in mA3 levels in these two strains has been attributed to the insertion of a xenotropic MuLV LTR near the 3' end of exon2 in the opposite orientation within the mA3 genes of B6, NZB, and RIIIS, but not in BALB/c, AKR, and C3H/He mice [72]. In addition, LPS binding to TLR4 leads to signaling through MyD88 [73] to yield increased AID transcripts in splenocytes from both B6 and

BALB/c strains [18,74], although AID protein levels are higher in BALB/c splenocytes, particularly after 48 hr of IL-4/LPS treatment (Fig 7). Previous reports have shown that viruses, including MMTV, use LPS on bacterial cells to facilitate infection [75,76], yet this signal likely will increase cytidine deaminase levels that curtail virus replication, necessitating Rem degradation of AID. However, Rem expression does not lead to mA3 degradation [19], and Rem antagonism of Apobecs is not demonstrable after TBLV infection of B6 mice, which have high constitutive mA3 levels relative to AID.

AID is known to alter gene expression epigenetically [77]. Hematopoietic precursor/stem cells (HPSCs) lacking AID expression on the B6 background upregulate the expression of multiple genes, including those encoding the transcription factors Cepbα [78], Klf1 [79], and Gata1 [80]. However, decreased gene expression is observed for suppressor of cytokine signaling 3 (Socs3) [81]. *Aicda*^-/- HPSCs showed skewing of hematopoiesis toward the myeloid lineage [77]. Interestingly, our RNA-seq analysis revealed that *Socs3* mRNA was increased in the absence of both AID and Rem expression (Fig 8), in addition to altered mRNA expression of IFN-induced genes and those involved in growth factor and cytokine signaling pathways. These data suggest that AID has effects on innate immunity and cytokine signaling and that Rem expression manipulates such pathways.

Why don't proviral hypermutations correlate with TBLV-induced tumor latency in B6 mice? One explanation is that Apobec-mediated hypermutations reduce replication in T lymphocytes, but recombination with endogenous *Mtv*s allows selection for replication-competent TBLVs that cause insertional mutagenesis [19]. Such viral recombinants have been observed in both MMTV-induced mammary tumors and TBLV-induced T-cell lymphomas [19,82,83]. Another non-exclusive possibility is that the multiple gene products of Rem have separate effects on proviral mutation and tumor susceptibility. As suggested previously, uncleaved Rem is likely an AID antagonist [27], whereas SP is a Rev-like protein that facilitates MMTV mRNA export and expression [21,25,84]. In contrast, the Rem C-terminal cleavage product, Rem-CT, has an unusual trafficking pathway through the ER and endosomes [27]. SP is synthesized by both TBLV-WT and SD, suggesting that differences in gene expression between these viruses are due to uncleaved Rem and Rem-CT. Since uncleaved Rem co-expression likely leads to AID proteasomal degradation [19], we speculate that Rem-CT is responsible for suppressing tumor latency, perhaps through cytokine manipulation.

## Materials and methods

### Ethics statement

The University of Texas at Austin (UTA) Institutional Biosafety Committee approved all work with retroviruses under BSL-2 or ABSL-2 conditions. Mouse experiments were performed with approval by the UTA Institutional Animal Care and Use Committee.

### Mouse infections

BALB/cJ and C57BL/6J mice were obtained from Jackson Laboratory (Bar Harbor, ME). *Aicda*^-/- mice on the B6 background were generated in the laboratory of Dr. Tasuku Honjo [18] and were kindly provided by Dr. Michel Nussenzweig (The Rockefeller University). The C57BL/6J mice deficient in both mA3 and AID (mA3/AID DKO) were derived by mating *Aicda*^-/- mice, a generous gift from Dr. James Hagman (National Jewish Health) and initially prepared by Dr. Tasuku Honjo [18], with mA3^-/- mice, a kind gift from Dr. Warner Greene (Gladstone Institute) [85]. Both strains had been backcrossed to C57BL/6J for at least 9 generations. F1 progeny from *Aicda*^-/- and mA3^-/- mice were crossed with each other to generate mA3/AID double-knockout mice.

The μMT mice on the B6 background were purchased from Jackson Laboratory (stock #002288). Mice were engineered by insertion of a neomycin-resistance cassette into the immunoglobulin mu heavy chain locus to interrupt the transmembrane domain [43]. This mutation prevents the cell surface expression of IgM and innate, naturally occurring antibodies, but mutant mice also have increased levels of plasmacytoid dendritic cells [45]. For each strain of mice, age-matched weanlings 4-to-6 weeks of age were injected intraperitoneally with either 1 x $10^6$ (low dose) or 2 x $10^7$ (high dose) stably transfected Jurkat T cells. The infectious clones of TBLV-WT and TBLV-SD in a plasmid vector expressing hygromycin resistance have previously been described [39]. Infectious clones were used for transfection of Jurkat cells and selected for three weeks in hygromycin and then screened for production of equivalent amounts of TBLV-WT or TBLV-SD as judged by Western blotting with antibodies to capsid protein (p27) [57]. Injected mice were monitored for the development of thymic and/or splenic tumors for a period of 9–12 months. T-cell lymphomas were detected by their restriction of lung function/breathing of the animals. Mice were sacrificed immediately since they die quickly (12 to 24 h) after these symptoms are observed. Large differences in the size and weight of tumors were not observed.

## DNA isolation, cloning, and Sanger sequencing

Genomic DNA extracted from tumors induced by TBLV-WT or TBLV-SD was used for PCR with primers *env*7254(+) (5'-ATC GCC TTT AAG AAG GAC GCC TTC T-3') and LTR9604 (-) (5'-GGA AAC CAC TTG TCT CAC ATC-3') for the region spanning the *env*-LTR junction. PCRs were performed in 25 μl with JumpStart RED Accutaq LA polymerase (Sigma-Aldrich, cat# D8045) in the supplied buffer, 500 ng of tumor DNA, 25 to 50 pmol of each primer, and 0.5 mM deoxynucleoside triphosphates. PCR parameters were: 94˚C for 1 min, 10 cycles at 94˚C for 10 sec, 53˚C for 30 sec, 68˚C for 2 min followed by 25 cycles of 95˚C for 15 sec, 50˚C for 30 sec, and 68˚C for 2 min with a final incubation at 68˚C for 7 min. Sequences encompassing the SD site and the *env* gene were acquired after cloning the PCR fragments inserted into pGEM-Teasy (Promega). Sanger sequencing with primers env7254(+) and env8506(-) (5'-GCA CTT GGT CAA GGC TCC TCG-3') was used to verify clones as previously described [19]. Sanger sequencing enabled identification of proviruses containing recombinants with endogenous *Mtv* proviruses.

## PCRs and RT-PCRs

To analyze the numbers of LTR-enhancer repeats, PCR was performed with 10 pmol of primers TBLV-LTR408(+) (5'- CCA ATA AGA CCA ATC CAA TAG GTA GAC -3') and TBLV-LTR786(-) (5'- CAC TCA GAG CTC AGA TCA GAA C -3'), 100 or 200 ng of tumor DNA, and JumpStart Taq ReadyMix (Sigma, cat# P0982). PCR parameters were: 94˚C for 5 min, then 35 cycles at 94˚C for 30 sec, 56˚C for 30 sec, 72˚C for 30 sec, and a final incubation at 72˚C for 7 min. Semi-quantitative PCR was performed with *Mtvr2* as the single copy gene standard using primers *Mtvr2*(+) (5'-TCT GGG ATC CGC TTC CTC AT-3') and *Mtvr2*(-) (5'-CCA GTC CTT GGC CCT CAT TTA-3'). MMTV-specific primers *pol*4235(+) and *pol*5835(-) in the viral polymerase gene were used to measure proviral sequences [19].

The qRT-PCRs were performed in triplicate using a ViiA7 Real-time PCR System (Applied Biosystems). Briefly, 1 μg of total DNase I-treated RNA was reverse transcribed using the Superscript first-strand synthesis kit (Invitrogen). The cDNA was diluted in DNase-RNase free water, and 20 ng was used for the reaction. The expression of the *Gapdh* gene was used for normalization. The reaction mixture consisted of 20 ng cDNA, 2.5 μM of each forward and reverse primer, and 7.5 μl of 2x Power SYBR Green in 15 μl. The primers used were: Stat1

forward 5'-TAC GGA AAA GCA AGC GTA ATC T-3' and reverse 5'-TGC ACA TGA CTT GAT CCT TCA C-3'; Socs3 forward 5'-ATG GTC ACC CAC AGC AAG TTT-3' and reverse 5'-TCC AGT AGA ATC CGC TCT CCT-3'; Gapdh forward 5'-GTG TGA ACG GAT TTG GCC GTA-3' and reverse 5'-GGA GTC ATA CTG GAA CAT GTA G-3'. The ΔΔCt method [86] was used for quantitating relative gene expression.

## High throughput sequencing and analysis

DNA extracted from tumors induced by TBLV-WT and TBLV-SD at the higher dose (3 tumors each) in wild-type and *Aicda*[-/-] B6 mice was used for PCR with primers *env*7254(+) (5'-ATC GCC TTT AAG AAG GAC GCC TTC T-3') and LTR9604(-) (5'-GGA AAC CAC TTG TCT CAC ATC-3'), resulting in an ~2 kb amplicon of the *env*-LTR junction region. Reactions were performed with JumpStart RED Accutaq LA polymerase (Sigma-Aldrich) in the supplied buffer, 1 μg of tumor DNA, 50 pmol of each primer, and 0.5 mM deoxynucleoside triphosphates in 20 μl. PCR parameters were: 94˚C for 1 min, then 10 cycles at 94˚C for 10 sec, 53˚C for 30 sec, and 68˚C for 2 min followed by 25 cycles of 95˚C for 15 sec, 50˚C for 30 sec, and 68˚C for 2 min with a final incubation at 68˚C for 7 min. Sixteen independent reactions for each tumor sample were pooled, and DNA was sheared to a size of approximately 400 bp using a Covaris ultrasonicator at the UT Austin Genomic Sequencing and Analysis Facility (GSAF). Fragmented DNA was end repaired and used for dA-tailing and ligation with Illumina adapters (unique to each tumor sample). Adapter-ligated DNA was amplified using these PCR parameters: 98˚C for 30 sec, 6 cycles at 98˚C for 10 sec, 65˚C for 30 sec, and 72˚C for 30 sec with a final incubation at 72˚C for 5 min. Library preparations were submitted for sequencing on the MiSeq platform at GSAF.

For the high-throughput sequence analysis of TBLV proviruses from the tumors induced by low-dose infection, the same PCR fragment used for Sanger sequencing was purified using a Zymoclean Gel DNA Recovery Kit (Zymo Research, cat# D4002). DNA (~1 μg) was prepared for Nanopore long amplicon sequencing by the University of Wisconsin Biotechnology Center. After eliminating likely PCR duplicates and quantifying the number of reads aligned with each base at each position relative to the reference sequence, alternative base frequencies were dichotomized using a cutoff of 3%. For each pair (reference base to alternate base), the dichotomized alternate calls were modeled as a function of sample group (TBLV-SD versus TBLV-WT in wild-type, AID-knockout, or mA3/AID double-knockout B6 mice). A mixed-effects logistic regression approach was used, incorporating a random effect that accounted for sample variation within the group.

## RNA extraction and RNA-seq analysis

Frozen thymic tumor tissue was ground to a fine powder with a mortar and pestle and lysed in TRI Reagent Solution (ThermoFisher Scientific, cat#AM9738) using a microtube-sized Dounce homogenizer. Total RNA was prepared using the Direct-zol RNA miniprep kit (Zymo Research #R2050) following the manufacturer's protocol. Total RNA (1.25 μg) was used for RNA Tag-Seq analysis at the UT Austin GSAF. Genes with significant differential expression (p<0.05) in the TBLV-SD-induced tumors were compared to those induced by TBLV-WT in the *Aicda*[-/-] background and subjected to volcano plots and PANTHER pathway analysis (www.pantherdb.org). All significant differences in mRNA frequencies are reported in S1 and S2 Tables.

## Virion isolation from tumors and RT-PCR

Each thymic tumor sample (~0.1 g) was homogenized in 0.75 ml of buffer containing 10 mM Tris-HCl, pH 8.0, 0.1 M NaCl, and 1 mM EDTA. Samples were subjected to centrifugation at 600xg for 5 min. The supernatant then was further clarified at 12,400xg for 10 min to remove debris. Virions were concentrated by ultracentrifugation at 105,000xg for 1 h, and the viral pellet was resuspended in 0.2 ml of PBS containing 0.5 M sucrose. Virion RNA was extracted using the GeneJET viral DNA and RNA Purification Kit (Thermo Scientific, cat# K0821). Purified virion RNA was treated with DNase I (Thermo Scientific, cat# EN0521) to remove any contaminating DNA, and the RNA was quantitated. Complementary DNA (cDNA) was prepared using SuperScript First Strand Synthesis System for RT-PCR (Thermo Scientific, cat#11904108) using 1 μg of virion RNA in a 20 μl reaction. A portion of the cDNA reaction (10%) was used for PCRs using the same *env*-LTR primers described for proviral DNA from tumors. Products were subjected to Nanopore sequencing and bioinformatic analysis.

## Splenocyte isolation and treatments

Spleens were removed aseptically and crushed to make a single-cell suspension in 5 ml of a solution of PBS plus 1% fetal bovine serum (FBS). Cells were subjected to centrifugation for 7 min at 335Xg and resuspended in 3 ml of red blood lysis solution (9 parts 0.15 M $NH_4Cl$ plus 1 part 0.15 M Tris-HCl, pH 7.6). Following a 5-min incubation at room temperature, 20 ml of PBS/FBS solution was added, and the cells were pelleted again. The pellet was resuspended in 4 ml of PBS-FBS solution, filtered through a 40 μM cell strainer, and counted for viable cells using trypan blue.

For some experiments, $1 \times 10^7$ splenocytes were added to 100mm Petri dishes containing 15 ml of stimulation media [RPMI 1640 (Sigma R8758) containing antibiotic-antimycotic solution (Gibco 15240062), 1 mM sodium pyruvate (Sigma P5280), 10% FBS, 50 μM 2-mercaptoethanol, 5 ng/ml IL-4 (Sigma I1020), and 20 μg/ml LPS (Sigma L-2630)]. In other experiments, $3 \times 10^7$ splenocytes were added to 100mm Petri dishes containing 15 ml of complete RPMI media (RPMI 1640, 10% FBS, 100 U/ml penicillin, 100 μg/ml streptomycin, 2 mM L-glutamine, 50 μg/ml gentamycin sulfate) supplemented with 4 to 5 μg/ml ConA (Thermo Scientific AAJ61221MC). To determine the effects of type I interferons, $1 \times 10^7$ cells were added to 100mm Petri dishes containing 15 ml of complete RPMI media plus 200 to 1000 U/ml IFN-β (R&D Systems 8234MB010). All cells were grown in a 37˚C incubator with 7.5% $CO_2$ for 4 h to 4 days (as indicated) prior to cell extract preparation.

## Cell lysate preparation and Western blotting

Whole cell extracts were prepared by adding either one volume of 2X-SDS-loading buffer [250 mM Tris-HCl, pH 6.8, 20% glycerol, 2% sodium dodecyl sulfate (SDS), 5% β-mercaptoethanol, 0.1% bromophenol blue] to cells in one volume of phosphate-buffered saline (PBS) or one volume of 6X-SDS loading buffer (0.35 M Tris-HCl, pH 6.8, 10% SDS, 36% glycerol, 0.6 M DTT, 0.012% bromophenol blue) to cells in five volumes of PBS. Samples were boiled for 5 min for protein denaturation. Protein concentrations were determined by Bradford assay (Bio-Rad Protein Assay Dye Reagent Concentrate, cat#5000006). The same amounts of whole cell lysates (20–25 μg) were loaded on 12% polyacrylamide gels and transferred onto nitrocellulose membranes (Cytiva Amersham Protran Supported NC Nitrocellulose Membranes, cat#10600037). After blocking with Intercept Blocking Buffer (LI-COR, cat#927–60001), blots were incubated with the primary antibody overnight at 4˚C. The secondary antibody was incubated with membranes for 1.5 h at room temperature. After each antibody incubation, blots were washed three times for 5 min at room temperature in TBS-T (0.1% Tween-20, 20 mM Tris, 136 mM

NaCl, pH 7.6). Bands were detected by LI-COR scanning or chemiluminescence (Amersham cat# RPN2232).

Proteins were extracted from ~0.1 g of each thymic tumor using EZLys Tissue Protein Extraction Reagent (Abcam, cat# ab286872) according to manufacturer's instructions. The protein concentration was determined, and equal amounts of lysates were used for Western blotting.

Antibodies to AID, p-Stat1(Y701), and Gapdh were obtained from Cell Signaling Technology. The mA3 antibody was kindly provided by Drs. Leonard Evans and Stefano Boi (NIAID Rocky Mountain Laboratories) [66,67]. Horseradish peroxidase-linked secondary antibodies were obtained from Cell Signaling Technology, whereas IRDye-linked secondary antibodies were obtained from LI-COR Biosciences.

## Statistical analysis and reproducibility

Results of Kaplan-Meier plots were analyzed for significance by Mantel-Cox log-rank tests. Statistical differences in the numbers of mutations/clone were evaluated by non-parametric Mann-Whitney tests. A p-value of $<0.05$ was considered to be significant. All experiments were repeated at least twice with similar results.

## Supporting information

**S1 Fig. Expression of mA3 in low-dose TBLV-induced T-cell lymphomas. (A) Representative Western blot analysis for mA3 levels in individual tumors.** Total protein was extracted from individual T-cell tumors induced by TBLV-WT or TBLV-SD and subjected to Western blotting with antibodies specific for mA3 or Gapdh. Four tumor-derived protein lysates from each virus/mouse strain combination were used for analysis. Results are shown for two B6 tumors induced by TBLV-WT (lanes 1 and 2), two B6 tumors induced by TBLV-SD (lanes 3 and 4), one B6 *mA3/Aicda*-double knockout tumor induced by TBLV-WT (lane 5), two B6 *Aicda*[-/-] tumors induced by TBLV-WT (lanes 6 and 7), and two B6 *Aicda*[-/-] tumors induced by TBLV-SD (lanes 8 and 9). A faint background band is observed in tumor extracts from mice lacking mA3 expression (lane 5). (B) Quantitation of mA3 levels in TBLV-induced T-cell tumors. The mA3 levels in each of four B6 tumors induced by TBLV-WT (black dots) were quantitated by LI-COR software relative to Gapdh, and the mean was assigned a relative value of 1.0 (horizontal line). The vertical bars show the standard deviations from the means. The mA3 levels of four independent TBLV-SD-induced B6 tumors (red squares), TBLV-WT-induced B6 *Aicda*[-/-] tumors (green triangles), and TBLV-SD-induced B6 *Aicda*[-/-] tumors (blue inverted triangles) are shown. Means for each group (horizontal lines) relative to the mean of values from four TBLV-WT-induced B6 tumors are shown by horizontal lines. Standard deviations from the means are shown by vertical bars. No statistical differences were observed. (TIF)

**S1 Table. mRNA sequences that have significantly different frequency between tumors induced by TBLV-WT (five tumors) or TBLV-SD (four tumors) in wild-type B6 mice.** (PDF)

**S2 Table. mRNA sequences that have significantly different frequency between tumors induced by TBLV-WT (five tumors) or TBLV-SD (five tumors) in *Aicda*[-/-] B6 mice.** (PDF)

## Acknowledgments

We thank the Nussenzweig, Hagman, and Greene laboratories for kindly providing knockout mice on the B6 background. The generosity of Dr. Leonard (Pug) Evan's laboratory in providing the mA3-specific antibody prior to his untimely death was greatly appreciated. We also valued the advice of Dr. Stefano Boi.

## Author Contributions

**Conceptualization:** Hyewon Byun, Gurvani B. Singh, Wendy Kaichun Xu, Jaquelin P. Dudley.

**Data curation:** Anna Battenhouse, Dennis C. Wylie, Jaquelin P. Dudley.

**Formal analysis:** Hyewon Byun, Wendy Kaichun Xu, Poulami Das, Anna Battenhouse, Dennis C. Wylie, Jaquelin P. Dudley.

**Funding acquisition:** Jaquelin P. Dudley.

**Investigation:** Hyewon Byun, Gurvani B. Singh, Wendy Kaichun Xu, Alejandro Reyes, Mary M. Lozano.

**Methodology:** Poulami Das.

**Project administration:** Jaquelin P. Dudley.

**Resources:** Mario L. Santiago.

**Supervision:** Hyewon Byun, Mary M. Lozano, Jaquelin P. Dudley.

**Writing – original draft:** Mary M. Lozano, Jaquelin P. Dudley.

**Writing – review & editing:** Hyewon Byun, Gurvani B. Singh, Wendy Kaichun Xu, Poulami Das, Anna Battenhouse, Dennis C. Wylie, Mario L. Santiago, Mary M. Lozano, Jaquelin P. Dudley.

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
