## [Decision Letter · Decision Letter 0]

14 Dec 2023

Dear Dr. Dudley,

Thank you very much for submitting your manuscript "Apobec-Mediated Retroviral Hypermutation In Vivo is Dependent on Mouse Strain" for consideration at PLOS Pathogens. As with all papers reviewed by the journal, your manuscript was reviewed by members of the editorial board and by 3 independent reviewers. In light of the reviews (below this email), we would like to invite the resubmission of a significantly-revised version that takes into account the reviewers' comments. Reviewers 1 and 2 had several concerns about the data and interpretation that need to be addressed., which may include providing additional data. Reviewer 3 felt that addition of more explicit description about the differences detected between the various mouse strains should be provided, as well as a better discussion of the differences in the Apobec and AID genes (and proteins) between BALB and B6 mice.

We cannot make any decision about publication until we have seen the revised manuscript and your response to the reviewers' comments. Your revised manuscript is also likely to be sent to reviewers for further evaluation.

Sincerely,

Susan R. Ross, PhD

Section Editor

PLOS Pathogens

Susan Ross

Section Editor

PLOS Pathogens

Kasturi Haldar

Editor-in-Chief

PLOS Pathogens

orcid.org/0000-0001-5065-158X

Michael Malim

Editor-in-Chief

PLOS Pathogens

orcid.org/0000-0002-7699-2064

Reviewer's Responses to Questions

**Part I - Summary**

Reviewer #1: In this manuscript, Byun et al examine the role of Rem in vivo using a superantigen independent lymphoma inducing strain of MMTV (TBLV). While previously they examined the role of MMTV in BALB/c mice, in this manuscript they utilize C57BL/6 mice. They show that unlike BALB/c mice, B6 mice are more susceptible to TBLV infection and tumorigenesis and the absence of Rem accelerated tumorigenesis in these mice. However, unlike their previous findings in BALB/c mice, they found that in C57BL/6 mice Rem did not affect the hypermutation levels. Finally, RNA seq data provided by the authors show that there are increased transcripts related to growth factor and cytokine signaling in mice infected with a TBLV-Rem deficient virus. The paper is well written and the conclusions are mostly justified based on the data presented. However, the paper lacks mechanistic details in 1) what is the deaminase responsible for the phenotype or 2) how is Rem affecting the expression of the genes described in tables S1 and S2.

Reviewer #2: In this manuscript, Byun et al. followed up their previous study on the role of Rem in the induction of T-cell lymphomas with TBLV and compared tempos of tumor development and patterns of proviral DNA mutations in B6-based genetically modified mouse strains. This is a potentially interesting study but the present manuscript is limited to be an assortment of preliminary data that need to be refined with more careful experiments. Although the title of the paper indicates the discovery of strain-depended differences in Apobec-mediated hypermutation, most of the data shown on patterns of hypermutation are negative or only indicate the effect of B-cell deficiency. The only possible strain-dependent difference is shown in Figure 7, which is inadequate.

Reviewer #3: This study evaluates the roles of Apobec proteins AID and mA3 in mutagenesis of pathogenic mouse mammary tumor viruses and consequent tumorigenesis. This work is an extension of an analysis done in BALB mice (ref 18) but in this case uses the more widely used C57B6 (B6) mice, and relies exclusively on the MMTV TBLV strain because B6 mice have an MHC mutation that fails to present the viral Sag protein, which is necessary for amplification of MMTV infection. The analysis includes AID null B6 mice and describes tumor latency, mutation types, expression of AID and mA3 and immune related transcripts. Specific comments:

1 – The comparison of similar experiments in BALB and B6 is interesting and the key impetus for doing these experiments, so the authors should consider adding a summary table outlining the observed similarities and differences.

2 – Some of the strain differences could be due to differences in the AID and mA3 genes in these mice. The authors do not specifically point out the known functional differences between BALB and B6 mA3 although some of these differences are mentioned here and there. These genes differ in sequence (not mentioned), expression levels (Fig 7), and splicing (fig 7). It would help to include a summary of these mA3 differences and also say something about any differences in AID genes. Line 326 should include a reference to work showing mA3 levels are enhanced in B6 relative to BALB.

3 – The authors repeatedly state that the differences in viral mutations suggest that the mutations at different motifs are not produced by the same enzymes (lines 223, 247, for example), but it is not clear if the authors just mean AID vs mA3 or are suggesting involvement of some other gene or cofactor. IOW do the authors think that mutations at ATC are AID induced rather than by mA3?

4 – Abstract. The summary but not the abstract points out that GtoA and CtoT mutations are diagnostic of Apobecs. Also, mMT mice are not defined here or in the text.

5 – Fig 1B. It is not absolutely clear which of the diagrams are TBLV-SD. Maybe the Xs could be defined in a key as “ X: TBLV-SD mutations”, and the term TBLV-SD removed from its current position. Also, it is odd to term the triplication in the LTR “flanking sequences” as they do not seem to flank anything as drawn. It might also be helpful to add arrows for the primers used to amplify the env/LTR segment used for sequencing.

6 – Fig 8c. Cytokine signaling is included in the list of differentially expressed genes, but is not included in the chart although Line 359 specifically sites cytokine signaling genes as involved in Rem/AID interactions.

7 – line 207. Definition of WRC should be in line 205.

**Part II – Major Issues: Key Experiments Required for Acceptance**

Reviewer #1: 1. The authors are looking at proviral DNA levels in tumors. What about the amount of virus present in each tumor? They should purify virus from the tumors and measure viral RNA levels.

2. When looking at the tumors from the different strains in Figure 2. Where they similar in size or weight?

3. In lines 148-149, the authors state that “the absence of AID eliminated the modest effect of Rem on TBLV-induced T cell tumors in B6 mice”. While that would be correct if indeed there had been a difference in 2A, unfortunately the p value was only 0.076 (>0.05). Thus, it is not statistically significant in 2A and this conclusion is not supported.

4. Figures 2D-F are really not well laid out. So, the third column in all three is labeled as “C57BL/6” and does that represent the three diploid copies of endogenous Mtvs in B6? However, when looking at TBLV-SD proviral DNA levels in D or E they look very similar to those in C57BL/6 collumn. Does that mean that the tumors do not contain any virus and all that is being “picked up” is endogenous Mtv?

5. For Table 1, 2 and 3, while the data are interesting, it is not clear why the authors are not also looking at mutations in RNA purified from virions from tumors from the different mice. Also, they should perform infectivity assays with the purified virions to determine how infectious the virus is.

6. Furthermore, when looking at the different mutations in tables 1,2 and 3. Did you observe any hot spots? Where there common sites mutated in the provirus DNA from the different tumors? (even from the different mouse strains?)

7. The authors state in line 238 “AID is not the sole enzyme responsible for provirus mutation”. This is evident from the data presented. However, it is not clear why the authors in figure 2, did not include B6 mAPOBEC3 knockout mice, or why they did not also use mAPOBEC3/Aicda double knockouts. That would strengthen this argument.

8. In figure 3, the authors give a breakdown of all the mutations identified per clone. It is always that these mutations are because of mutating host enzymes. But, what about the role of reverse transcriptase. RT has no proofreading activity and induces a lot of mutations. How do the authors distinguish mutations from host editing enzymes vs RT?

9. In figure 7a and lines 315-322, the authors are making comparisons in the intensity of the bands from their western blot. The authors must do densitometry analysis of the different proteins under the different conditions to support the statements in lines 315-322. Also, how many times have the experiments, shown in Figure 7, been done? This information is not present at the figure legend.

10. The authors in figure 7 show the effect of different compounds on the induction of mA3 and mAID. How is infection affecting the protein levels of mAID and mA3? This information is very important for the manuscript. What about the mAID and mA3 protein levels in tumors? Are they elevated, lower?

11. Lines 341-344, the authors discuss the relative abundance of transcripts after TBLV infection (WT vs Rem knockout) in the supplemental tables between B6 and B6Aicda-/-. However, what about in BALB/c? Do they see similar trends in transcripts?

12. Why are the authors in Figure 8D are looking at Stat1, when Stat4 was the one that was affected? Is this a typo or not?

13. In line 381-382, the authors state that the reduced dose produced similar tumor incidence btw BALB/c and B6 mice. Why is that? The authors should provide a reason why they see that in the Discussion.

14. Line 388-389, could not be though another mechanism? What about difference in sites of integration between TBLV wt and Rem knockout. The authors never looked if there are differences in the integration site patterns between these two viruses that could explain differences in tumor induction.

15. In lines 395-396, the authors state that because the proviral loads are lower in Aicda-/- compared to B6 and uMT, this suggests that mature B cells inhibit TBLV replication. However, this is confusingly written. Shouldn’t then B6 mice be similar to the Aicda-/- mice? B6 have mature B cells and yet their proviral loads are similar to uMT (at least with the wild type virus). This part needs to be rewritten, because it is not clear. Are the authors comparing both viruses (WT and Rem knockout) or just Rem knockout?

16. In lines 406-407, the authors state that the mA3 levels are low in immature B cells. They should look at that and provide the data, as it will strengthen their manuscript.

Reviewer #2: 1) Data shown in Figure 7, especially the high expression level of AID in Con A-stimulated BALB/c spleen cells, might be very important but the experimental procedures must be refined. The authors stimulated spleen cells with Con A, IFNbeta, or IL-4 + LPS for 4 days. However, Apobec protein expression can be induced at much earlier time-points (Tsukimoto et al. Viruses 14:832, 2022). Time course of protein expression must be analyzed first. Further, although Con A is relatively selective in stimulating T cells, Con A-activated T cells produce multiple cytokines, which may in turn activate other immune cell types to express AID. Thus, at least purified T and B cells must be used to conclude that BALB/c T cells express AID. In this regard, why did not the authors include muMT mouse spleen cells in this experiment? Further, if there is a strain-dependent difference in T-cell expression of AID, some mechanistic analyses must be performed. Could this be because of the possible polymorphisms in promoter sequence that result in binding of different nuclear factors?

2) Bands in Figure 6, panel B are smeary and unclear. Are tumors in AID-KO mice polyclonal and some cells contain more than 5 copies of LTR enhancer?

3) In Figures 1 and 4, Kaplan-Meier plots are shown for "tumor-free" fraction at different time-points. However, it is unclear how "tumor-free" status was defined and how many animals were examined for each plot. In the Materials and Methods, line 496, the authors state that mice were monitored for the development of thymic and/or splenic tumors, but the legend for the same figure states that mice were followed for tumor development including enlarged thymus, spleen, and lymph nodes. While the spleen is palpable and lymphadenopathies in the axillar and/or inguinal regions can be seen, thymic enlargement and lymphadenopathies in the abdominal cavity is difficult to observe from the outside. Did the authors inoculated a large number of animals and killed some of them periodically for pathological examination? More detailed methods for detecting tumors must be described. In this regard, in lines 907-908 the authors describe "survival plots." Are these actually survival curves? As the number of animals followed to generate each plot is unclear, it cannot be judged if the barely significant difference between TBLV-WT- and TBLV-SD-infected groups in Figure 2C might actually depend on only one survivor.

Reviewer #3: Points one and two in my comments could be considered as being more important than the others. No new experiments were requested.

**Part III – Minor Issues: Editorial and Data Presentation Modifications**

Reviewer #1: 1. Please briefly describe in the Abstract and material and methods what uMT mice are to enhance clarity.

2. Line 181 “these data are consistent…..TBLV replication”. Is that based on a previous report, “this idea”? If yes, please provide the reference to it.

3. Lines 223-225. What are the authors trying to say here. It is not clearly written. What figures are they comparing? In what sense was it “dramatically different”?

4. In Figure 3, how many sequences were obtained from each tumor and each strain of mice?

5. A typo in line 259. Shouldn’t it be p<.0001 and not p<.001?

6. In lines 284 and 287. The authors state that they analyzed tumors from high dose infection and refer to figure 5B. However, Figure 5B states in the figure legend that it is low dose. Which one is right?

7. In line 409-410, why didn’t the authors observe a difference in uMT tumors? They should provide a hypothesis about it.

Reviewer #2: 1) Horizontal and vertical axes do not start with zero in Figure 2, panels A-C, while they do start with zero in Figure 4. Vertical axis in Figure 2, panel E does not match with the relative expression levels indicated on the top of each bar. How does the authors explain the same tempo of tumor development in the absence of increased proviral load shown in Figure 2, panels D-F? Is promoter insertion not involved in the induction of T-cell tumors?

2) The authors repeatedly describe conclusive remarks without enough logical context, for example in lines 148, 174-175, 181. In lines 230-231, "Aicda-insufficient (why not deficient?) mice infected with TBLV-SD relative to those in TBLV-SD-infetcted mice" does not make sense. How the authors can conclude that the mutations occur after infection of immature B cells in lines 249-250? Similarly, what does "TBLV-SD-induced tumors are accelerated" mean in line 258? Does IFNbeta "simulate a viral infection" as described in line 312? If so, how?

3) The text is filled with unusual and mistaken expressions. G-to-A mutations "typical of Apobecs" must be mutations typical of those induced by Apobecs. AID is activation-induced cytidine deaminase, not activation-stimulated. Cytidines are not mutated, but are substituted. What does "slighter greater" mean in line 317? The entire manuscript must be proofread for proper English usage.

Reviewer #3: Points 3-7 in my comments

PLOS authors have the option to publish the peer review history of their article (what does this mean?). If published, this will include your full peer review and any attached files.

Reviewer #1: No

Reviewer #2: No

Reviewer #3: No
---

## [Decision Letter · Decision Letter 1]

29 Apr 2024

Dear Dr. Dudley,

Thank you very much for submitting your manuscript "Apobec-Mediated Retroviral Hypermutation In Vivo is Dependent on Mouse Strain" for consideration at PLOS Pathogens. As with all papers reviewed by the journal, your manuscript was reviewed by members of the editorial board and by  2 of the previous reviewers. In light of the reviews (below this email), we would like to invite the resubmission of a significantly-revised version that takes into account the reviewers' comments.

Specifically, both reviewers felt that some of their previous significant concerns had still not been addressed. This includes verifying expression levels of APOBEC3 and AID in different cells examined in the study and clarification of the stages of B cells examined in the study.

We cannot make any decision about publication until we have seen the revised manuscript and your response to the reviewers' comments. Your revised manuscript is also likely to be sent to reviewers for further evaluation.

Sincerely,

Susan R. Ross, PhD

Section Editor

PLOS Pathogens

Susan Ross

Section Editor

PLOS Pathogens

Michael Malim

Editor-in-Chief

PLOS Pathogens

orcid.org/0000-0002-7699-2064

Reviewer's Responses to Questions

**Part I - Summary**

Reviewer #1: In this manuscript, the authors examine the role of Rem in vivo using TBLV, a Sag independent lymphoma inducing MMTV strain. The authors examine the effect of TBLV on B6 mice in the presence or absence of Rem. Their main goal is to understand the role of Rem and its effect on mA3 and AID in the context of different mouse lines (B6 vs BALB/c). The authors have been partially responsive to the comments of this reviewer. A number of important questions have not been answered by the authors and their answer will strengthen the manuscript significantly. For example, looking at deaminase protein levels in tumors.

Reviewer #2: This revised manuscript reads easier than the previous version and the authors' logic is better understood with the rewritten text. However, the results are still largely descriptive and no mechanistic analyses were performed. Further, some descriptions are left uncorrected from the previous version and more careful interpretation of the presented data is required in view of the previously published results.

**Part II – Major Issues: Key Experiments Required for Acceptance**

Reviewer #1: 1. Comment 5 of prior review has not been fully addressed. This reviewer understands that there is not a good infectivity assay for virus from TBLV tumors. However, what about mutations in viral RNAs purified from tumors? That has not been addressed and it would add to the significance of the findings.

2. The authors must show mA3 and mAID protein levels in tumors, as they are looking at mutation rates in viral DNA purified from tumors in this manuscript. The correlation of the protein levels of these deaminases and viral DNA mutation levels must be examined.

3. Regarding comment 16, the authors assume that mA3 is higher in immature B cells. This assumption is based on incomplete data and the authors should do this experiment to address this extrapolation.

Reviewer #2: 1. Relatively large amounts of data are presented without logical connections between each other.　What are possible explanations for higher levels of viral DNA for Rem-negative TBLV-SD infection in muMT mice? Is this possibly associated with the lack of mature B cells in muMT mice and a resultant lack of antiviral antibody production? AID deficiency does not affect the formation of retrovirus-neutralizing antibodies in mice. Levels of viremia may affect the multiplicity of viral replication cycles and resultant mutation rates. The results described in the "lack of Rem expression accelerates WT-B6 and AID-deficient tumors after low-dose inoculation" section on page 11 collectively indicate that Rem interferes with TBLV-induced tumor development independently of AID, tumors derived from low-dose inoculation showed the most abundant poviral mutations in the TYC context typical of mA3, and significantly lower levels of G-to-A mutations were observed when AID- and mA3- double knockout mice were infected with either TBLV-WT or TBLV-SD. These observations indicated that it would be mA3 that is involved in the mutagenesis of proviruses in B6 mice. Nevertheless, the possible tumor induction was not examined in mA3-KO mice and nothing is discussed about this obvious conclusion in the Discussion section. Why do the authors ignore the possible importance of mA3 when it is known that mA3 is expressed in high levels in B6 mice? Even without showing any mechanistic data, the authors should provide readers with some interpretations of presented results that lead to future mechanistic analyses.

2. Figure 7 is still problematic. There are vertical lines between lanes 96h and NS for B6 in panel A and also between NS and 4h lanes for BALB in panel B, indicating that these NS blots are from separate lanes. Are they from a single gel or were electrophoresis and blotting performed separately? Even if they were from a single experiment, the bands for AID in panel A are blurry, and the abrupt induction of AID protein at 96 hours after stimulation in BALB cells alone is highly surprising. Previous reports have shown the induction of AID mRNA in B6 B cells at 24 hours after LPS stimulation alone or with combined LPS and anti-mu stimulation (Kuraoka, M. et al. Cell Reports 18: 1627-1635,2017). AID protein expression of B6-derived B cells at 48 hours after LPS + IL-4 stimulation has also been shown (Pone, E.J. et al. J. Immunol. 194: 3065-3078, 2015). Further, the lack of AID protein expression at 96 hours after LPS + IL-4 stimulation is inconsistent with the authors' own Figure S1. As to Con A stimulation the authors conclude that mA3 expression was observed after 4 hours in BALB/c spleen cells but this was delayed in B6 mice. However, compared to the line-separated NS lane, mA3 protein levels increased at 4 hours but then decreased at 18 hours after stimulation, while those in B6 mice decreased once at 4 hours and increased after 48 hours. Why are these complex changes? Moreover, although the authors describe "changes in AID levels as shown in Fig. 7B and Fig. S1," no data on AID is shown in Figure 7B, and Figure S1 shows expression levels at 4 days after stimulation alone with no changes. While the authors describe that the experiments were performed at least three times, the arrangement of lanes is different between panels A and B, which can be easily corrected in repeated experiments.

3. The authors use the term "immature B cells" in an inappropriate meaning. In immunology, immature B cells are those at a specific stage of B-cell development at which cells have undergone negative selection in the bone marrow and express surface IgM, but not IgD. These immature B cells then migrate to the spleen and differentiate through IgMHi, IgD- T1 and IgMHi, IgDMid T2 stages to finally reach the IgMLo, IgDHi mature follicular B cell stage. The authors seem to describe B cell precursors with the term "immature B cells" as surface IgM+ immature B cells do not develop in muMT mice since B cell development is blocked before the preB stage in these animals. Thus, their descriptions in lines 249-251 on mutations occurring after infection of "immature B cells" in muMT mice do not make sense. Similarly, B6 mice do not express MHC class II E molecules due to a lack of Ealpha chain expression. A lack of Sag presentation is not due to a mutation.

4. In the Discussion section, references are cited arbitrarily. For example, inhibition of MuLV replication by mA3 was shown in ref. 55 as well as in cited ref.65-67. In lines 482-484, the authors cite ref. 73 in the context of increased AID levels in splenocytes, but ref. 73 only describes the involvement of MyD88 in LPS-induced B cell activation and AID expression was not examined.

**Part III – Minor Issues: Editorial and Data Presentation Modifications**

Reviewer #1: No minor issues.

Reviewer #2: a. In line 36 muMT mice do not "lack the ability to make IgM." They can synthesize the mu heavy chain, but its expression on the surface of B cell precursors is blocked due to a lack of the transmembrane domain.

b. In line 39 "proviral G-to-A or C-to-T proviral mutations" is repetitive.

c. In lines 43-44 "TBLV-SD-induced tumors relative to those from TBLV-WT" does not make sense. Do the authors mean tumors from TBLV-WT-infected mice?

d. The authors appropriately describe AID as activation-induced cytidine deaminase in lines 31 and 56, but in lines 90-91, they describe it as activation-associated.

e. The sentence in lines 106-107 starting with "the Apobec-induced hypermutation phenotype of this mutant" is unclear in its meaning, partly because of the presence of both mutation and mutant. MMTV-SD lacks the expression of Rem and hypermutation is observed in the absence of Rem. What the authors seem to argue here is that as SP is commonly expressed between wild-type and SD viruses, it must be the C-terminal sequence of Rem that is responsible for the lack of hypermutation in its presence.

f. What does "most transcriptional changes were observed in tumors from Aicda-/- mice" mean?

g. In line 142, most wild-type B6 mice inoculated with TBLV-WT, not any B6 strains tested.

h. In lines 153-154, again muMT is not immunoglobulin heavy chain knockout.

i. In line 160, the latency is compared between tumors, and thus not significantly different from "that in wild-type B6 mice."

j. The descriptions on TLV-SD are inconsistent. In the Introduction just TBLV-SD is used, and in the Results TBLV-SD (Rem-null) is additionally used in line 241, while in the Discussion TBLV-SD (Rem-null) is repeatedly used.

k. In line 236, the second TBLV-SD must be the wild-type virus.

l. In line 268, tumors cannot be accelerated, tumor development is.

m. In line 273 tumor latency between TBLV-WT and SD does not make sense. Tumor latency differs between TBLV-WT and TBLV-SD infections.

n. In line 274 "the proviral region spanning the env-LTR region" is repetitive.

o. In lines 280-281, we cannot conclude that the cytidine mutations are likely due to a different enzyme, but can consider it to be due to an enzyme other than AID. What actual enzymes do the authors have in mind here?

p. In line 332, "uncultured" is misleading as IL-4/LPS stimulation is performed in cell culture.

PLOS authors have the option to publish the peer review history of their article (what does this mean?). If published, this will include your full peer review and any attached files.

Reviewer #1: No

Reviewer #2: No
---

## [Decision Letter · Decision Letter 2]

14 Aug 2024

Dear Dr. Dudley,

We are pleased to inform you that your manuscript 'Apobec-Mediated Retroviral Hypermutation In Vivo is Dependent on Mouse Strain' has been provisionally accepted for publication in PLOS Pathogens. Please note that 1 reviewer asked for re-wording of a sentence. I believe that this can be addressed as the manuscript undergoes processing.

Best regards,

Susan R. Ross, PhD

Section Editor

PLOS Pathogens

Susan Ross

Section Editor

PLOS Pathogens

Michael Malim

Editor-in-Chief

PLOS Pathogens

orcid.org/0000-0002-7699-2064

Reviewer Comments (if any, and for reference):

Reviewer's Responses to Questions

**Part I - Summary**

Reviewer #1: The authors have taken many steps to address the issues identified by the reviewers by providing mA3 levels in tumors as well as determining G to A mutations in viral RNA. The Discussion has been revised to be more reflective of the work presented in the manuscript. Overall, this paper has been strengthened significantly.

Reviewer #2: The manuscript and figures have been modified with additional experiments, as this reviewer requested. The text is still difficult to comprehend, but the revised Discussion helps to understand the authors' logic.

**Part II – Major Issues: Key Experiments Required for Acceptance**

Reviewer #1: No issues any longer.

Reviewer #2: This reviewer found no major issues.

**Part III – Minor Issues: Editorial and Data Presentation Modifications**

Reviewer #1: (No Response)

Reviewer #2: In lines 415-416 "single or double-knockout Apobec strains" does not make sense. Please rephrase.

In line 676 RPMI 1640 is a culture medium and thus "stimulation media containing RPMI 1640" does not make sense.

PLOS authors have the option to publish the peer review history of their article (what does this mean?). If published, this will include your full peer review and any attached files.

Reviewer #1: No

Reviewer #2: No

---

## [Editor Report · Acceptance letter]

22 Aug 2024

Dear Dr. Dudley,

We are delighted to inform you that your manuscript, " Apobec-Mediated Retroviral Hypermutation In Vivo is Dependent on Mouse Strain ," has been formally accepted for publication in PLOS Pathogens.

Best regards,

Michael Malim

Editor-in-Chief

PLOS Pathogens

orcid.org/0000-0002-7699-2064